# HLA-A*11:01-restricted CD8⁺ T cell immunity against influenza A and influenza B viruses in Indigenous and non-Indigenous people

Jennifer R. Habel[1ↀ], Andrea T. Nguyen[2,3ↀ], Louise C. Rowntree[1], Christopher Szeto[2,3], Nicole A. Mifsud[2], E. Bridie Clemens[1], Liyen Loh[1], Weisan Chen[3], Steve Rockman[1,4], Jane Nelson[5], Jane Davies[5], Adrian Miller[6], Steven Y. C. Tong[5,7], Jamie Rossjohn[2,8], Stephanie Gras[2,3‡], Anthony W. Purcell[2‡], Luca Hensen[1,9‡], Katherine Kedzierska[1‡*], Patricia T. Illing[2‡]

1 Department of Microbiology and Immunology, University of Melbourne, at the Peter Doherty Institute for Infection and Immunity, Parkville, Australia, 2 Infection and Immunity Program & Department of Biochemistry and Molecular Biology, Biomedicine Discovery Institute, Monash University, Clayton, Australia, 3 La Trobe Institute for Molecular Science, Department of Biochemistry and Chemistry, La Trobe University, Bundoora, Australia, 4 Seqirus, Parkville, Australia, 5 Menzies School of Health Research, Darwin, Australia, 6 Indigenous Engagement, CQUniversity, Townsville, Australia, 7 Victorian Infectious Diseases Service, The Royal Melbourne Hospital, and Doherty Department University of Melbourne, at the Peter Doherty Institute for Infection and Immunity, Parkville, Australia, 8 Institute of Infection and Immunity, Cardiff University, School of Medicine, Cardiff, United Kingdom, 9 Department of Internal Medicine, Hematology, Oncology, Clinical Immunology and Rheumatology, University Hospital Tübingen, Tübingen, Germany

ↀ These authors contributed equally to this work.
‡ SG, AWP, LH, KK and PTI also contributed equally to this work.
* kkedz@unimelb.edu.au

**Data Availability Statement:** The mass spectrometry HLA-A*11:01 immunopeptidome data sets have been deposited to the

## Abstract

HLA-A*11:01 is one of the most prevalent human leukocyte antigens (HLAs), especially in East Asian and Oceanian populations. It is also highly expressed in Indigenous people who are at high risk of severe influenza disease. As CD8⁺ T cells can provide broadly cross-reactive immunity to distinct influenza strains and subtypes, including influenza A, B and C viruses, understanding CD8⁺ T cell immunity to influenza viruses across prominent HLA types is needed to rationally design a universal influenza vaccine and generate protective immunity especially for high-risk populations. As only a handful of HLA-A*11:01-restricted CD8⁺ T cell epitopes have been described for influenza A viruses (IAVs) and epitopes for influenza B viruses (IBVs) were still unknown, we embarked on an epitope discovery study to define a CD8⁺ T cell landscape for HLA-A*11:01-expressing Indigenous and non-Indigenous Australian people. Using mass-spectrometry, we identified IAV- and IBV-derived peptides presented by HLA-A*11:01 during infection. 79 IAV and 57 IBV peptides were subsequently screened for immunogenicity in vitro with peripheral blood mononuclear cells from HLA-A*11:01-expressing Indigenous and non-Indigenous Australian donors. CD8⁺ T cell immunogenicity screening revealed two immunogenic IAV epitopes (A11/PB2₃₂₀₋₃₃₁ and A11/PB2₃₂₃₋₃₃₁) and the first HLA-A*11:01-restricted IBV epitopes (A11/M₄₁₋₄₉, A11/NS1₁₈₆₋₁₉₅ and A11/NP₅₁₁₋₅₂₀). The immunogenic IAV- and IBV-derived peptides were >90% conserved among their respective influenza viruses. Identification of novel immunogenic HLA-A*11:01-restricted CD8⁺ T cell epitopes has implications for understanding how

ProteomeXchange Consortium via the PRIDE [47] partner repository with the dataset identifier PXD028985 and 10.6019/PXD028985. The final crystal structure models for the HLA-A*11:01 complexes have been deposited to the Protein DataBank (PDB) under the following accession codes: HLA-A*11:01- NP511-520: 7S8Q, HLA-A*11:01- M141-49: 7S8R and HLA-A*11:01-NS1186-195: 7S8S. The remaining data are fully available without restriction. All the remaining data are within the manuscript and its Supporting Information files.

**Funding:** This study was supported by the Australian National Health and Medical Research Council (NHMRC) Program Grant (#1071916) to K. K., NHMRC Project Grant (#1122524) to K.K., S.Y. C.T., A.M., S.G. and A.W.P., NHMRC Investigator Grant (#1173871) to K.K., the Research Grants Council of the Hong Kong Special Administrative Region, China (#T11-712/19-N) to K.K, NHMRC Project grant (#1085018) to A.W.P. and N.A.M. Salary support: K.K. was supported by the NHMRC Investigator Fellowship (#1173871), J.R. by an ARC Laureate fellowship (FL160100049), S.G. by an NHMRC SRF-A Fellow (#1159272), A.W.P. by an NHMRC Principal Research Fellowship (#1137739) and S.Y.C.T. is an NHMRC Career Development Fellow (#1145033). E.B.C. was supported by an NHMRC Peter Doherty Fellowship (#1091516), A.T.N. is supported by a Monash BDI PhD scholarship and an AINSE Ltd. Postgraduate Research Award (PGRA), J.R.H. was supported by the Melbourne Research Scholarship, L.H. was a recipient of Melbourne International Research Scholarship and Melbourne International Fee Remission Scholarship. The funders had no role in study design, data collection and analysis, decision to publish, or preparation of the manuscript.

**Competing interests:** I have read the journal's policy and the authors of this manuscript have the following competing interests: S.R. is an employee of Seqirus Ltd and has no conflict of interest in the material presented. The other authors declare no conflicts of interest.

CD8+ T cell immunity is generated towards IAVs and IBVs. These findings can inform the development of rationally designed, broadly cross-reactive influenza vaccines to ensure protection from severe influenza disease in HLA-A*11:01-expressing individuals.

## Author summary

Influenza A and influenza B viral infections cause significant morbidity and mortality. Established CD8+ T cell immunity directed at conserved viral regions provides protection against influenza viruses, drives rapid recovery, and leads to less severe clinical outcomes. Killer CD8+ T cells recognising viral peptides presented by HLA class I glycoproteins can provide broad immunity across distinct influenza strains and subtypes. Using immuno-peptidomics, we identified novel CD8+ T cell targets for influenza A and influenza B viruses in the context of HLA-A*11:01, an HLA-I allomorph highly prevalent in East Asia and Oceania, including Indigenous populations. Our study provides key insights for T cell-directed vaccines and immunotherapies.

## Introduction

Influenza viruses remain an annual epidemic human pathogen despite progress in vaccine formulation and anti-viral therapies. Three types of influenza viruses infect humans, type A (IAV), B (IBV) and C (ICV). IAVs (H3N2; H1N1) and IBVs co-circulate to cause seasonal epidemics of mild, severe, or fatal respiratory disease, while ICV can cause severe disease in children [1–4]. A major hindrance to long-term vaccine effectiveness is influenza virus shift and drift, making annual re-formulation of the vaccine a requirement to maintain immunity, although protection is not guaranteed [1].

Much of the focus on influenza virus vaccine formulation has been on inducing humoral immunity, involving influenza-specific antibodies targeting viral surface glycoproteins to neutralize the virus and prevent infection. However, it is well established that cellular immunity, particularly cytotoxic CD8+ T cells, plays a vital role in the clearance of influenza virus infection [5–7]. A previous study involving patients infected with highly fatal H7N9 virus revealed that those who recovered from infection had greater numbers of H7N9-specific CD8+ T cells than those who succumbed to infection [8]. Therefore, CD8+ T cell-mediated responses towards influenza virus-infected cells need to be considered for generating protective immunity. CD8+ T cells can confer broad cross-reactivity across all IAVs, IBVs and ICVs [9], having key implications for the design of universal vaccines that do not require annual reformulation. Vaccines eliciting cross-reactive cytotoxic CD8+ T cells could reduce annual rates of IAV and IBV-induced morbidity and mortality as well as protect children from ICV. As current influenza vaccines do not promote cytotoxic T cell memory [10], it is important to understand how to elicit protective CD8+ T cell immunity against seasonal, pandemic and recently emerged IAVs and IBVs.

CD8+ T cells recognize peptides presented on HLA class I (HLA-I) molecules and can form long-lasting memory after being primed by an immunogenic epitope. HLA alleles are highly polymorphic, with some alleles having higher frequencies in certain ethnic groups. Variations in HLA alleles within the population lead to differential immune responses to influenza viruses. The HLA-A*11:01 allele is a member of the HLA-A*03 supertype and is prominent in many Asian and Oceanian populations, including Indigenous populations who are a high-risk

group for severe disease with influenza virus infection. Morbidity and mortality resulting from the 2009 H1N1 pandemic virus was higher in Indigenous populations globally, including Indigenous Australians [11], Native Americans and Alaskans [12], Pacific Islanders [13] and Māori [14]. To date, only a handful HLA-A*11:01-restricted CD8+ T cell epitopes have been reported for IAVs [15–21], and IBV epitopes are yet to be described.

Here, we report the first known HLA-A*11:01-restricted IBV epitopes and assess the quality of known and newly identified IAV CD8+ T cell epitopes restricted by HLA-A*11:01. We used mass-spectrometry to identify IAV- and IBV-derived peptides presented by HLA-A*11:01 during infection. A total of 79 IAV and 57 IBV peptides identified by mass spectrometry were screened for immunogenicity *in vitro* with peripheral blood mononuclear cells (PBMCs) from HLA-A*11:01-expressing Indigenous and non-Indigenous Australian donors. We revealed two immunogenic IAV epitopes (A11/PB2$_{320-331}$ and A11/PB2$_{323-331}$) and the first HLA-A*11:01-restricted IBV epitopes (A11/M$_{41-49}$, A11/NS1$_{186-195}$ and A11/NP$_{511-520}$), all of which were >90% conserved among their respective influenza viruses. Identification of novel immunogenic HLA-A*11:01-restricted CD8+ T cell epitopes is key for understanding how CD8+ T cell immunity is generated towards IAVs and IBVs. These findings provide insights towards rational design of broadly cross-reactive influenza vaccines to protect HLA-A*11:01-expressing individuals from severe influenza disease.

## Results

### Prevalence of HLA-A*11:01 in Asian and Oceanian populations

HLA-A*11:01 is one of the most prevalent HLAs, especially in East Asia and Oceania. Compared to the 7.3% global distribution of HLA-A*11:01, the detected frequencies of HLA-A*11:01 were the highest in South-East Asia (24.5%), Oceania (21.9%), South Asia (13.9%) and Australia (11.8%) (Fig 1A). Previous work by our group using the *Looking into InFluenza T cell immunity* (LIFT) cohort determined HLA-A*11:01 as one of the most prominent HLA-I alleles present in Indigenous Australians [22]. The Allele Frequency Net Database (allelefrequencies.net) also revealed high frequency of HLA-A*11:01 in other Indigenous and Asian populations. High HLA-A*11:01 prevalence was found in Papua New Guinea Madang people (63.6%), China Yunnan Hani (61.3%), Taiwan Hakka (40.0%), Pakistan Brahui (25.2%), Vietnam Hanoi Kinh (22.9%), Cape York Peninsula Aboriginal people (18%) and New Zealand Māori (16.7%). In contrast, the lowest frequency of HLA-A*11:01 was detected in Caucasian people from the USA and Australia (7% and 6.7%, respectively) (Fig 1A). HLA-A*11:01 prevalence in our LIFT cohort of Indigenous Australians was at 16.1% (Fig 1A). Despite such prominence of HLA-A*11:01 in East Asia and Indigenous people in Oceania, only a handful of CD8+ T cell epitopes have been previously identified for IAV and there are currently no epitopes identified for IBV.

HLA-I profiling of the LIFT cohort further demonstrated that HLA-A*11:01 is the third most prevalent (16.1%) HLA-A allele, after A*34:01 (29%) and A*24:02 (24%), in Indigenous Australians and the fourth most prevalent (10.1%) in non-Indigenous Australians, after A*02:01 (22.9%), A*01:01 (16.8%) and A*03:01 (14.3%) (Fig 1B). HLA-A*11:01 was commonly co-expressed with HLA-A*34:01 and HLA-A*24:02 in Indigenous Australians, and with HLA-A*02:01 in Indigenous and non-Indigenous Australians. The most prominent HLA-B and HLA-C alleles co-expressed with HLA-A*11:01 included B*13:01 (32.9%) and C*04:01 (38.2%) in Indigenous Australians, and HLA-B*35:01, B*40:01, B*44:02 and C*07:02 in non-Indigenous Australians (Fig 1C).

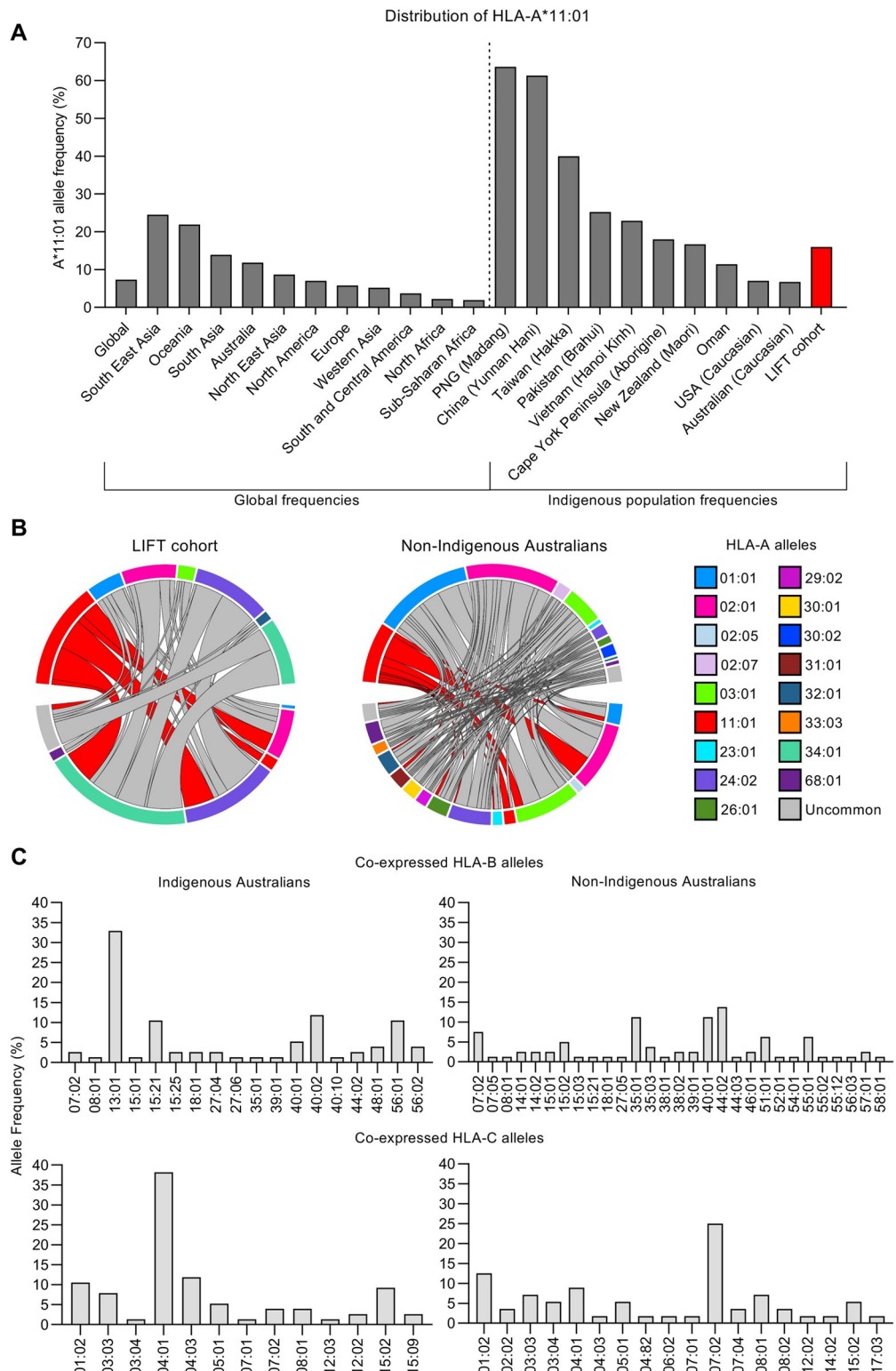

**Fig 1. Cohort demographics, HLA-A*11:01 population frequencies and HLA-I profiles of Indigenous Australians.**
(**A**) Frequency of HLA-A*11:01 according to geographic region (left panel) and within representative Asian (China, *n* = 150; Vietnam *n* = 170; Oman *n* = 118; Pakistan *n* = 104; Taiwan *n* = 55), Indigenous (Papua New Guinea (PNG) Madang, *n* = 65; Australia Cape York Peninsula Aborigine, *n* = 103), and Caucasian (Australia Caucasian, *n* = 134; USA Caucasian, *n* = 265) populations (right panel). Allele frequencies were obtained from the Allele Frequency Net

Database (allelefrequencies.net) HLA classical allele frequency search tool. (**B**) Circos plot displaying co-expression of HLA-A alleles within the Indigenous LIFT cohort (n = 127, alleles = 2n) and non-Indigenous cohort (n = 223, alleles = 2n). HLA-A*11:01 is depicted with red chords. (**C**) Frequency of HLA-B (top) and HLA-C (bottom) alleles co-expressed with HLA-A*11:01 in Indigenous LIFT cohort and non-Indigenous individuals (*n* = 38, alleles = 2n).

## CD8+ T cell responses towards known IAV-specific HLA-A*11:01-restricted epitopes

As a handful of HLA-A*11:01- and -A*03 supertype-restricted IAV peptides have been previously described [15–21], we first sought to determine which of those IAV-derived peptides could generate HLA-A*11:01-restricted CD8+ T cell responses. Using PBMCs from HLA-A*11:01-expressing individuals (S1 Table), HLA-A*11:01- and HLA-A*03 supertype-restricted IAV peptides were screened for immunogenicity in PBMCs expanded with two peptide pools encompassing these peptides (S2 Table). Overall, none of the previously reported IAV-derived peptides produced a significantly higher proportion of activated IFN-$\gamma^+$CD8+ T cells when compared to the DMSO negative control (Fig 2A). On an individual level, stimulation with $M1_{13-21}$ or $M1_{125-134}$ resulted in IFN-$\gamma$ CD8+ T cell responses in *n* = 3 or *n* = 1 donors, respectively. Since $M1_{125-134}$ was previously reported as an HLA-A*03 supertype-restricted IAV epitope [15], we further investigated $M1_{125-134}$-specific CD8+ T cell responses in HLA-A*11:01-expressing individuals. A systematic screening approach involving the 18-mer $M1_{121-138}$ and six overlapping 13-mer peptides spanning the $M1_{121-138}$ region, and the previously reported 10-mer $M1_{125-134}$, revealed that stimulation with $M1_{122-134}$, $M1_{124-136}$, and $M1_{126-138}$ peptides resulted in significantly higher frequencies of IFN-$\gamma^+$CD8+ T cells (Fig 2B). In addition, while not statistically significant, three donors (50% of donors, n = 6) also had IFN-$\gamma^+$CD8+ T cell responses to $M1_{123-135}$, $M1_{125-135}$, and $M1_{125-137}$ that were substantially higher than the DMSO negative control. Analysis of the shared amino acid sequences of the immunogenic 13-mer peptides allowed deduction of the minimal epitope to be A11/$M1_{126-134}$. To experimentally determine whether $M1_{126-134}$ was in fact the minimal epitope, PBMCs were stimulated and expanded with either $M1_{126-133}$, $M1_{126-134}$, $M1_{127-134}$, $M1_{125-134}$ or $M1_{121-133}$ as an irrelevant peptide control, followed by re-stimulation with each of the M1 peptide variants to measure IFN-$\gamma$ production. Indeed, $M1_{126-134}$ was the minimal peptide required in the $M1_{121-138}$ region to stimulate CD8+ T cell activation in HLA-A*11:01-expressing donors (50% of donors, *n* = 4), and the previously reported $M1_{125-134}$ elicited similar levels of CD8+ T cell activation (Fig 2C).

Epitope-specific CD8+ T cell polyfunctionality of responsive HLA-A*11:01-expressing individuals was assessed by analyzing simultaneous production of multiple cytokines and chemokines (IFN-$\gamma$, TNF, MIP1-$\beta$) and degranulation marker (CD107a). Analysis of CD8+ T cells expanded with $M1_{125-134}$ or $M1_{126-134}$ and re-stimulated with either $M1_{125-134}$ or $M1_{126-134}$ revealed similar proportions of cells with 1–4 functions, suggesting that stimulation with either variation of the peptide will induce CD8+ T cells with similar functionality (Fig 2D).

## Identification of IAV- and IBV-derived peptides presented by HLA-A*11:01 by immunopeptidome analysis

We have previously identified novel influenza epitopes using mass spectrometry-based analysis of the immunopeptidome for HLA-A*02:01 [9] and HLA-A*24:02 expressing C1R cells [23]. We thus utilized C1R cells transduced with HLA-A*11:01 to analyze the peptides presented by this allele during infection with either A/X31 or B/Malaysia/2506/04. HLA class I molecules were isolated using the pan class I antibody W6/32 and analyzed by mass spectrometry as described previously [24] to generate total of 7 HLA-A*11:01 immunopeptidome data

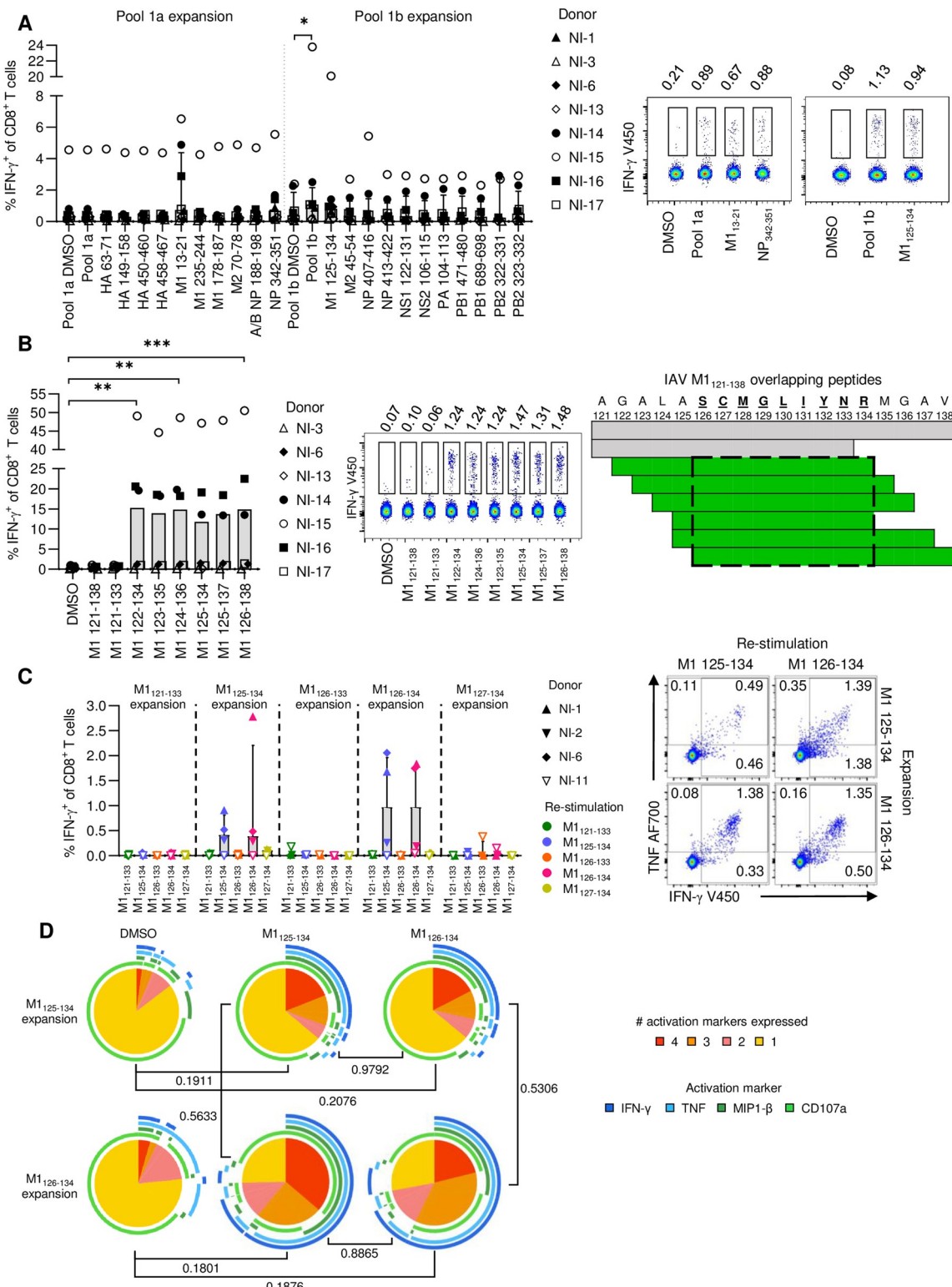

**Fig 2. CD8⁺ T cell responses towards known HLA-A*11:01-restricted IAV epitopes.** (**A**) Frequency of IFN-γ⁺ of CD8⁺ T cells stimulated with pooled and individual previously published HLA-A*11:01-restricted IAV epitopes. CD8⁺ T cells were expanded with either pool 1a or 1b. Median and IQR are shown ($n = 6–7$). Representative concatenated FACS plots of one donor are shown for IFN-γ responses to the negative control (DMSO), peptide pool 1a and 1b, M1$_{13-21}$, NP$_{342-351}$ and M1$_{125-134}$. (**B**) Frequency of IFN-γ⁺ of CD8⁺ T cells after stimulation with M1$_{121-138}$ and overlapping peptides within this region (n = 6–7). Representative concatenated FACS plots

of one donor are shown of the IFN-γ response to each condition. (**C**) Frequency of IFN-γ+ of CD8+ T cells after expansion and re-stimulation with variants of M1$_{126-134}$, with M1$_{121-133}$ as a negative control. Median and IQR are shown (*n* = 4). Representative FACS plots are shown for CD8+ T cell responses to M1$_{125-134}$ and M1$_{126-134}$ after expansion with either peptide. (**D**) Proportions of activated CD8+ T cells with 1–4 functions after expansion and re-stimulation with M1$_{125-134}$ or M1$_{126-134}$ (*n* = 4). A permutations test was used to determine statistical significance. (A-B) Friedman test with Dunn's correction for multiple comparisons was used to determine statistical significance.

sets: 3, 2 and 2 from uninfected, IAV (A/X31) infected and IBV (B/Malaysia/2506/04) infected C1R-A*11:01, respectively. These data sets were initially used to characterize the HLA-A*11:01 binding motif based on peptides assigned to the human proteome at a peptide spectrum match (PSM) false discovery rate (FDR) of 1% (described in *bioinformatic analysis of mass spectrometry data*). As C1R cells express endogenous HLA-B*35:03 (minimally expressed) and HLA-C*04:01, peptides observed in isolations from endogenous HLA-I (and HLA-II) data from previous studies [9,23] were firstly removed. After filtering, more than 21,000 non-redundant peptide sequences of 7–20 amino acids length were identified across the 7 data sets. As observed for HLA-A*11:01 binders [25], identified peptides were predominantly 9–11 amino acids in length (Fig 3A). Most frequent residues at position 2 of 9-11-mers were Val, Thr and Ser, while the positively charged Lys (and to a lesser extent Arg) dominated the C-terminal position (Fig 3B), as depicted on the IEBD (https://www.iedb.org/mhc/213).

A total of 159 peptides (non-redundant by sequence) were assigned to A/X31 at a 1% FDR. Although peptides assigned with scores below the 1% FDR threshold should be treated with increased caution without further validation, especially those with poor/lower scoring spectra, we also considered an additional 37 peptides identified at scores below this threshold and assessed their predicted binding to HLA-A*11:01 (S1 Dataset). Of the 196 A/X31 peptides, 58 and 34 were predicted to be strong (SB) and weak binders (WB) of HLA-A*11:01 respectively (using NetMHC4.0), while a further 38 were considered potential binders (PB) based on evidence of pull down and overlap with predicted binders. Similarly, a total of 109 peptides (non-redundant by sequence) were assigned to B/Malaysia/2506/04 at 1% FDR, with an additional 46 peptides identified having scores below this threshold. Of these, 27, 21 and 19 were predicted to be SB, WB or PB of HLA-A*11:01 respectively (S1 Dataset). These peptides were predominantly 9–11 amino acids in length, as for peptides assigned to the human proteome (Fig 3A, 3C and 3D). Most prominent proteins represented for A/X31 were PB2>PB1>NP, M1, while for B/Malaysia/2506/04 the hierarchy was HA, NP>M1, PB2 (Fig 3E, F). This is similar to previous analyses where A/X31 PB2 and PB1 peptides were prominent in the immunopeptidome of HLA-A*24:02, and B/Malaysia/2506/04 HA and NP peptides were prominent in the immunopeptidomes of HLA-A*02:01 and -A*24:02 [9, 23]. Due to the use of the pan-class I antibody W6/32 for HLA I isolation, influenza peptides with the capacity to bind HLA-B*35:03 and HLA-C*04:01 were also observed, as well as peptides with predicted capacity to bind more than one of the expressed HLA (S1 Dataset).

Of note, numerous potential alternative reading frame peptides mapping to the translation of the A/X31 genome alone were also assigned (Fig 3E and S1 Dataset). For simplicity, these peptides were named according to the frame starting at the start of the viral UTR, with positional numbering considering stop codons as equal to an amino acid (S1 Fig). The majority of these peptides were assigned with scores above the 1% FDR threshold (Fig 3E and S1 Dataset). Interestingly, two peptides mapping to the M2 intronic region of segment 7 (M(+3)$_{39-48}$ and M(+3)$_{94-103}$) are immediately preceded by potential alternative start codons within the RNA sequence (S1 Fig). Indeed, M(+3)$_{39-48}$ encompasses the reported alternative M2 ectodomain segment which forms the M42 protein [26].

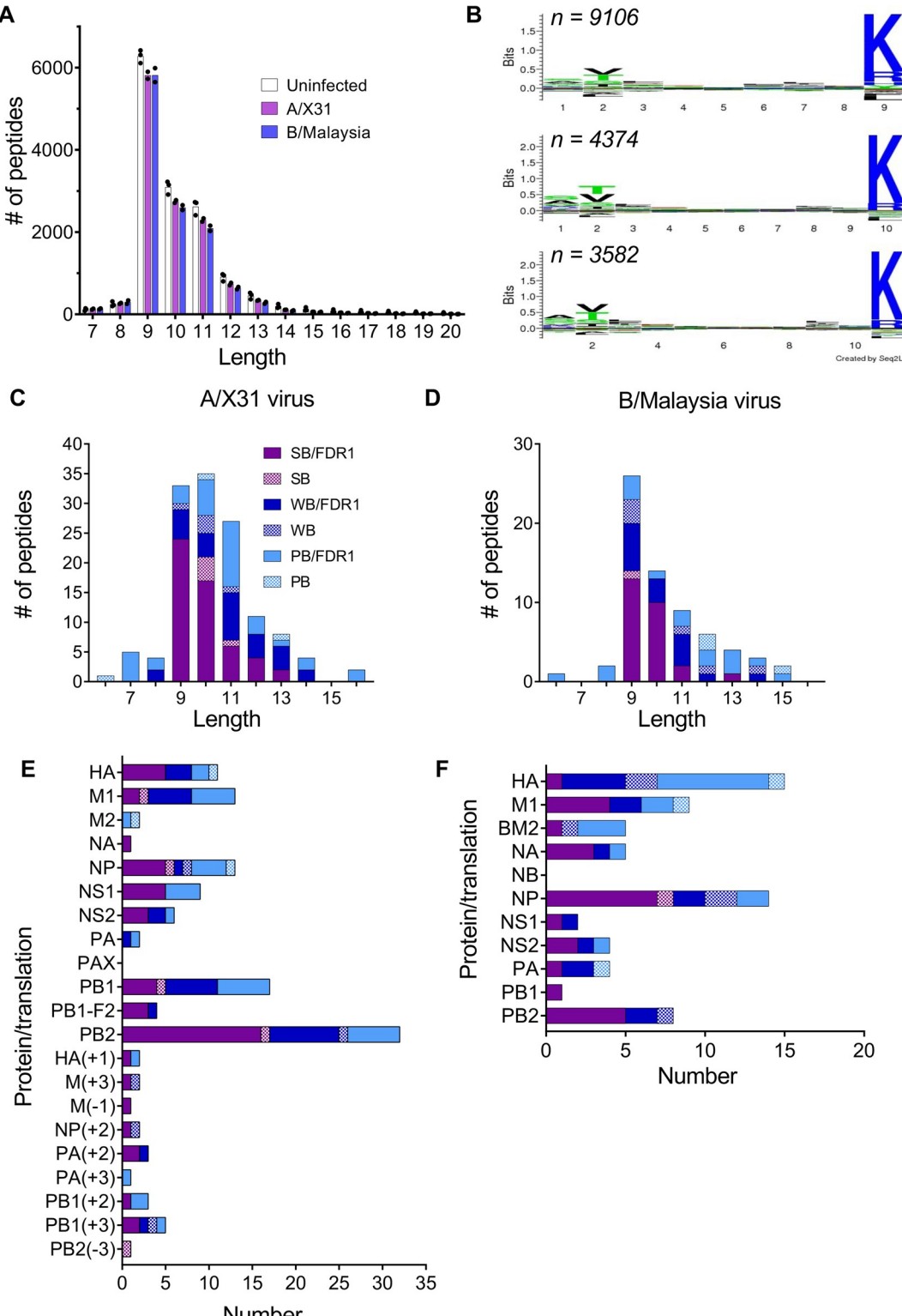

**Fig 3. Identification of novel HLA-A*11:01-restricted peptides for influenza A and influenza B viruses.** (**A**) Length distribution of HLA-A*11:01 ligands mapping to the human proteome (1% FDR, non-redundant by sequence), after filtering of peptides identified as ligands of endogenous HLA of the C1R cell line. Bars show the mean number of peptides of each length assigned in 3, 2 and 2 experiments for uninfected, A/X31-infected and B/Malaysia/2506/04-infected conditions, respectively. Points show peptide numbers for individual replicates. (**B**) Binding motif for 9–11 amino acid peptides generated from

combined non-redundant human sequences (1% FDR) across the seven data sets, where n is the total number of sequences for each length. Sequence logos were generated using Seq2logo 2.0 [46]. (**C, D**) Length and (**E, F**) proteome distributions of (**C, E**) A/X31- and (**D, F**) B/Malaysia/2506/04-derived ligands (non-redundant by sequence, combined from 2 data sets) identified as strong (SB, % Rank ≤ 0.5), weak (WB, 0.5 < % Rank ≤ 2) or potential binders (PB) of HLA-A*11:01 at a 1% FDR (solid bars) and at scores below the 1% FDR threshold (checked bars). %Rank binding calculated using NetMHC 4.0 [29,30]. Influenza peptide sequences are available in S1 Dataset.

Neither $M1_{126-134}$ nor any of the peptides containing this region were initially identified in A/X31 samples. Given this sequence contains a reactive Cys residue, searches of the C1R-A*11:01 datasets were repeated incorporating oxidations of Cys residues as possible peptide modifications. Both $M1_{125-134}$ and $M1_{126-134}$ peptides were identified, with oxidation at Cys and Met residues ($M1_{125-134}$ AS**CM**GLIYNR; $M1_{126-134}$ S**CM**GLIYNR) (S1 Dataset). The identification of $M1_{125-134}$ and $M1_{126-134}$ naturally presented by HLA-A*11:01 confirms epitope mapping (Fig 3B-D).

From these data, 79 A/X31 (including 33, 21, 19, and 6 predicted to be SB, WB, PB and non-binders (NB) respectively) and 57 B/Malaysia/2506/04 (25, 16, 6, and 8 predicted to be SB, WB, PB and NB respectively, 2 additional peptides >14 amino acids) were synthesised in their native forms for CD8+ T cell IAV and IBV epitope testing.

## Identification of novel IAV-specific HLA-A*11:01-restricted CD8+ T cell epitopes

To determine CD8+ T cell reactivity to the selected 79 IAV-derived peptides identified by LC-MS/MS, peptides were divided into 8 pools of increasing predicted binding affinity determined by the NetMHCpan 4.0 algorithm (S3 Table). CD8+ T cell lines were generated by stimulation with one of the 8 peptide pools for 10 days followed by re-stimulation with the respective pool to measure IFN-γ production. Frequencies of IFN-γ+CD8+ T cells were significantly higher than the respective DMSO negative control for pools A-C (Fig 4A). Dissection of individual peptides in pools A and B revealed that $PB2_{320-331}$ and $PB2_{323-331}$ were the most immunogenic, while none of the peptides in pool B resulted in significantly higher frequencies of IFN-γ+CD8+ T cells than background (Fig 4B and 4C). While stimulation with $PB1_{659-669}$ did not significantly increase CD8+ T cell activation, two donors had responses that were above the basal IFN-γ+CD8+ T cell frequencies observed in the negative control, suggesting this peptide may be a subdominant epitope within HLA-A*11:01 (Fig 4C). Stimulation with $PB2_{323-331}$, $PB2_{320-331}$, or $PB1_{659-669}$ did not induce strong polyfunctionality in responsive CD8+ T cells, with the majority of cells expressing one or two activation markers, namely MIP1-β and/or CD107a (Fig 4D).

Both PB2 epitopes were predicted strong binders of HLA-A*11:01 and the fragmentation spectrum of the synthetic peptides were well matched with the discovery spectrum of the eluted peptide (S1 Dataset). $PB2_{320-331}$ and $PB2_{323-331}$ are variants of a previously published HLA-A*11:01-restricted IAV epitope, $PB2_{322-331}$ [15,17] which did not elicit IFN-γ responses when tested using our CD8+ T cell probing method (Fig 2A). Although none of the alternative reading frame peptides tested were identified as major epitopes, fragmentation spectra of the synthetic peptides were consistent with the discovery spectra of the eluted peptides (S1 Dataset). Overall, our IAV CD8+ T cell epitope identification studies found alternative variants of a previously reported epitope $A11/PB2_{320-331}$ and $A11/PB2_{323-331}$, and confirmed their presentation on HLA-A*11:01 during IAV infection using mass spectrometry.

## Identification of novel IBV-specific HLA-A*11:01-restricted CD8+ T cell epitopes

IBV-derived CD8+ T cell epitopes restricted by HLA-A*11:01 have not been identified to date. While IBVs do not have the same pandemic potential as IAVs, they are clinically relevant and

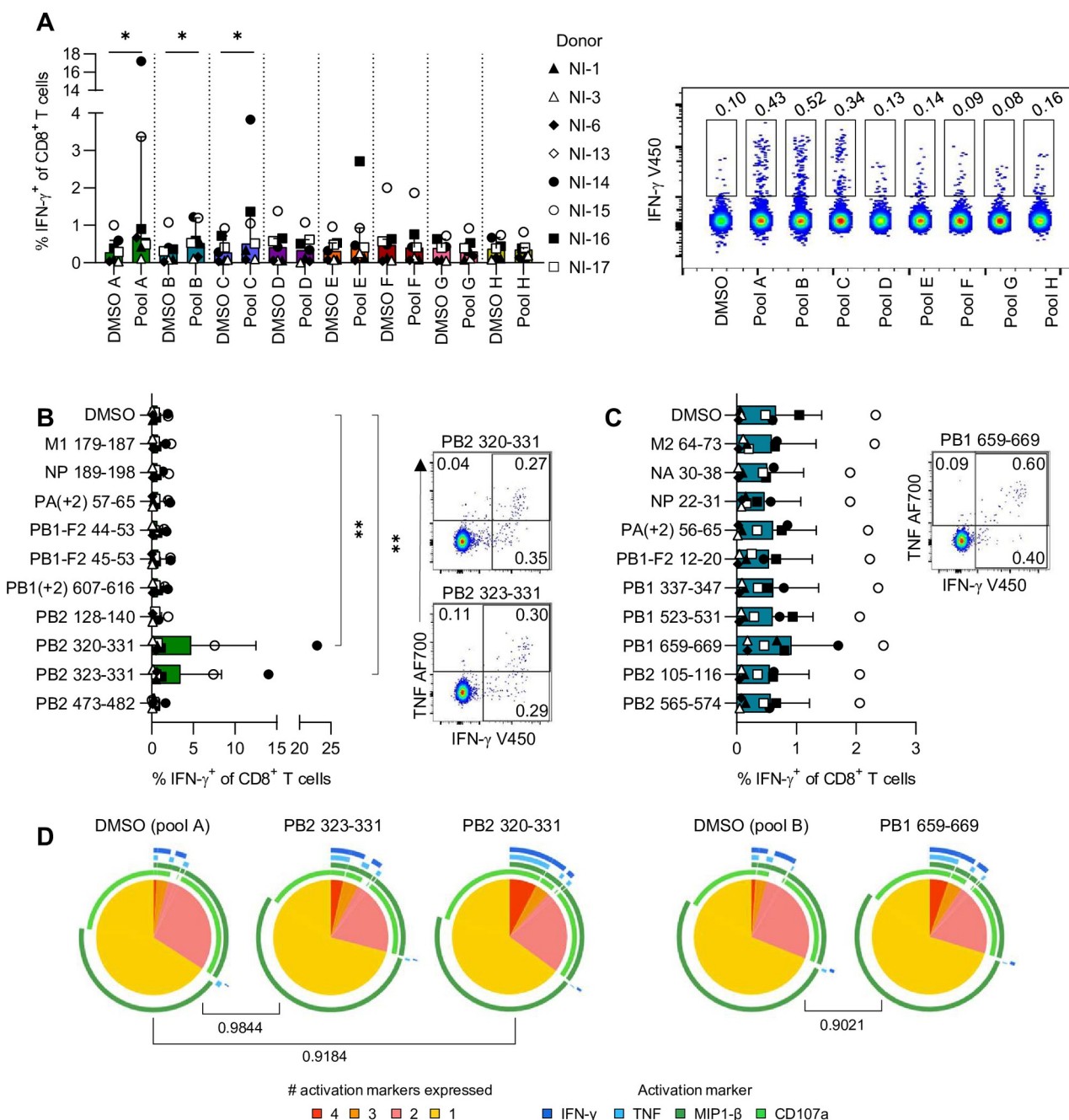

**Fig 4. CD8+ T cell reactivity towards IAV peptides.** (**A**) Frequency of IFN-γ+CD8+ T cells following stimulation with DMSO (negative control) and IAV peptide pools. Wilcoxon signed-rank test was used to determine statistical significance. Median and IQR are shown (*n* = 7). Representative concatenated FACS plots from one donor are shown. (**B**-**C**) Frequency of IFN-γ+CD8+ T cells after stimulation with individual IAV peptides from (**B**) Pool A and (**C**) Pool B. Friedman test with Dunn's correction for multiple comparisons was used to determine statistical significance. Median and IQR are shown (*n* = 6–7). Representative FACS plots showing CD8+ T cell responses to (**B**) PB2$_{320-331}$ and PB2$_{323-331}$, and (**C**) PB1$_{659-669}$. (**D**) Polyfunctionality of CD8+ T cells responsive towards PB2$_{323-331}$, PB2$_{320-331}$ or PB1$_{659-669}$, and the associated DMSO negative control. A permutations test was used to determine statistical significance.

can cause severe influenza disease, especially in young children [1,3]. To identify novel IBV-derived CD8+ T cell epitopes restricted HLA-A*11:01, we determined the immunogenicity of 57 LC-MS/MS-identified IBV peptides by probing memory CD8+ T cells in PBMCs of

HLA-A*11:01-expressing individuals. The 57 peptides were divided into three pools containing 19 peptides, each based on predicted binding affinity according to the NetMHCpan 4.0 algorithm (S4 Table). CD8+ T cell lines were generated by stimulating PBMCs with IBV-infected C1R-A*11:01 cells, which were expanded *in vitro* for 12 days. On day 13, PBMCs were re-stimulated with IBV peptide-pools to assess intracellular IFN-γ and TNF production.

Analysis of IFN-γ production by CD8+ T cells demonstrated that pools 1 and 2 were the main source of CD8+ T cell activation, as the frequencies of IFN-γ+CD8+ T cell were significantly higher than the DMSO negative control (Fig 5A). Individual peptides from pools 1 and 2 were then dissected to identify novel immunogenic IBV CD8+ T cell epitopes. Re-stimulation of PBMCs with individual IBV-derived peptides demonstrated that peptides in pool 1 did not produce robust CD8+ T cell responses, while $M1_{41-49}$ from pool 2 was the only peptide to elicit significantly higher IFN-γ+CD8+ T cell frequencies than the DMSO negative control (Fig 5B and 5C). While stimulation with $NP_{511-520}$ and $NS1_{186-195}$ did not result in significantly increased CD8+ T cell activation, some donors had lower magnitude responses that were still above the basal IFN-γ+CD8+ T cell frequencies observed in the negative control (n = 5 and n = 4, respectively), which suggests that these peptides form subdominant epitopes with HLA-A*11:01 (Fig 5B). CD8+ T cells stimulated with $M1_{41-49}$ and $NS1_{186-195}$ were found to have significantly higher proportions of polyfunctionality when compared to the negative control (Fig 5D). All three peptides were predicted to bind HLA-A*11:01 with an affinity stronger than 200nM (NetMHC4.0 and NetMHCpan 4.0, S4 Table and S1 Dataset) and the fragmentation spectrum of the synthetic peptides were consistent with the discovery spectrum of the eluted peptides (S1 Dataset).

Overall, our IBV CD8+ T cell epitope identification studies found novel $A11/M_{41-49}$, $A11/NS1_{186-195}$ and $A11/NP_{511-520}$ epitopes, representing the first reported CD8+ T cell epitopes for IBV restricted by the HLA-A*11:01 allomorph.

## HLA-A*11:01 complexes with IAV-derived PB2 peptides display low thermal stability

To understand the stability of HLA-A*11:01-restricted influenza CD8+ T cell epitopes, we assessed the thermostability of the IAV- ($PB1_{659-669}$, $PB2_{320-331}$ and $PB2_{323-331}$) and IBV- ($M_{41-49}$, $NS1_{186-195}$ and $NP_{511-520}$) derived peptides in complex with the HLA-A*11:01 molecule. HLA-A*11:01 heavy chain was refolded with each of IAV- ($PB2_{320-331}$, $PB2_{323-331}$ or $PB1_{659-669}$) or IBV-derived peptides ($NP_{511-520}$, $M1_{41-49}$ or $NS1_{186-195}$) and a thermal stability assay was performed using differential scanning fluorimetry. Four of the peptide-HLA (pHLA) complexes exhibited a thermal stability (Tm) of about 60˚C (S5 Table), in the range of other previously reported pHLA complexes [27]. The two overlapping PB2 peptides from IAV showed a decreased Tm value up to 10˚C lower (S5 Table).

## Structure of IBV-derived peptide presented by HLA-A*11:01 molecule

To better understand the molecular interactions of the identified epitopes, we sought to solve structures of the newly-identified IBV HLA-A*11:01-restricted CD8+ T cell epitopes. Structures comprising of HLA-A*11:01 in complex with the IBV peptides $NP_{511-520}$, $M1_{41-49}$ and $NS1_{186-195}$ were solved at a resolution of 2.08, 2.95 and 1.82 Å, respectively (S6 Table and S2 Fig). The 10-mer $NP_{511-520}$ peptide adopted a canonical conformation within the HLA-A*11:01 binding cleft, anchored by P2-Thr and P10-Lys (Fig 6A). P2-Thr forms a hydrogen bond with Glu63, while the side chain of P10-Lys is buried deep in the F pocket of HLA-A*11:01, forming a salt bridge with Asp116 and makes hydrophobic contacts with Ile95, Ile97 and Leu81 (S3A Fig). The exposed surface of the $NP_{511-520}$ peptide is relatively flat despite

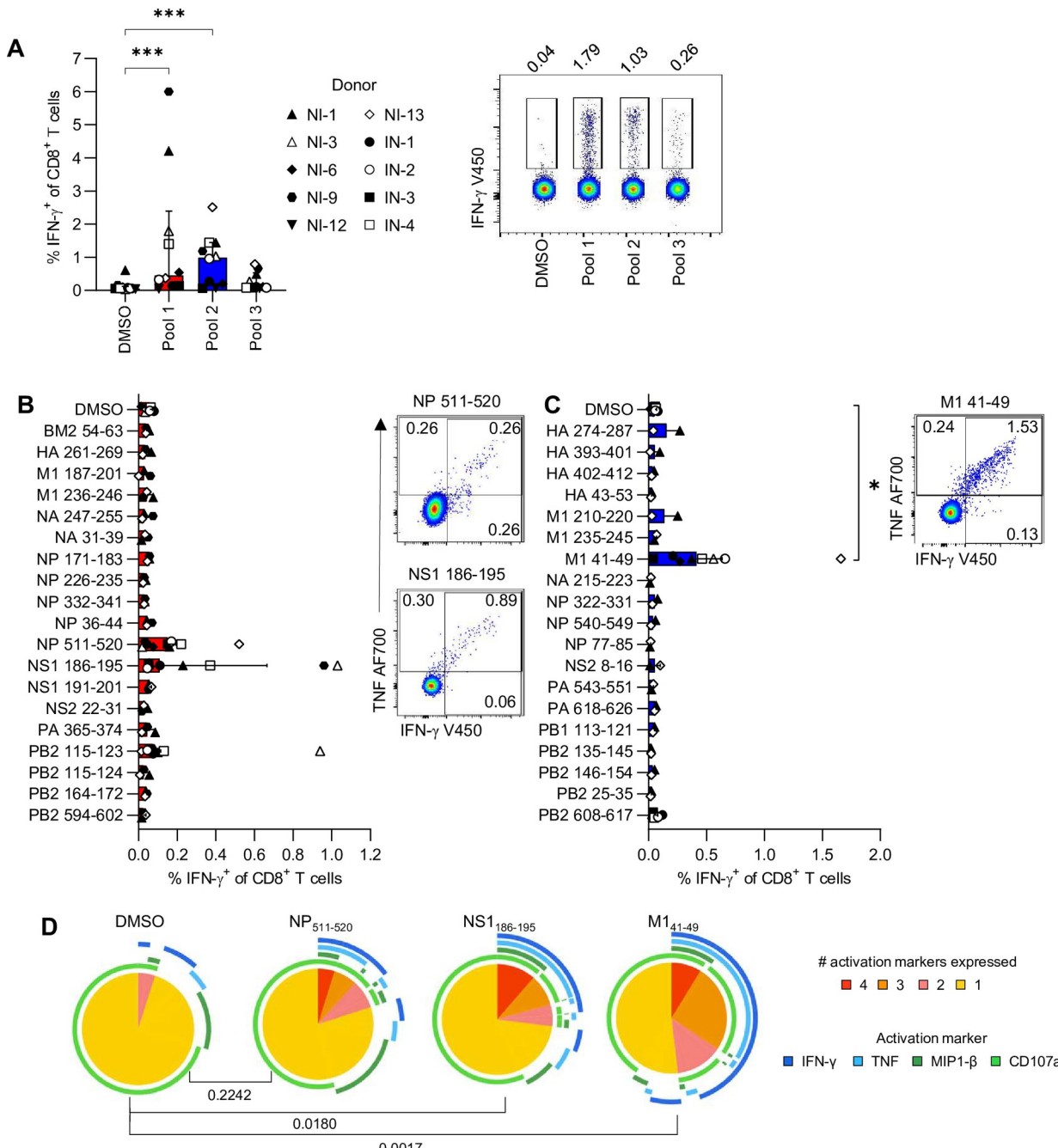

**Fig 5. CD8+ T cell responses to novel IBV epitopes.** (**A**) Frequency of IFN-γ+CD8+ T cells after stimulation with individual IBV peptide pools. Median and IQR are shown (*n* = 10). Representative FACS plots are shown for CD8+ T cell responses from one donor towards peptide pool 1–3 and the negative control. (**B**) Frequency of IFN-γ+CD8+ T cells in response to stimulation with individual peptides from Pool 1. Median and IQR are shown (*n* = 2–9). A representative FACS plot of one donor with a CD8+ T cell response to NP$_{511-520}$ or NS1$_{186-195}$. (**C**) Frequency of IFN-γ+CD8+ T cells in response to stimulation with individual peptides from Pool 2. Median and IQR are shown (*n* = 2–9). Representative FACS plot of one donor with a CD8+ T cell response to M1$_{41-49}$. (**D**) Proportions of CD8+ T cells with 1–4 functions after stimulation with NP$_{511-520}$, NS1$_{186-195}$ or M1$_{41-49}$ (*n* = 8 or 9, respectively). A permutations test was used to determine statistical significance. (**A**-**C**) Friedman test with Dunn's correction for multiple comparisons was used to determine statistical significance.

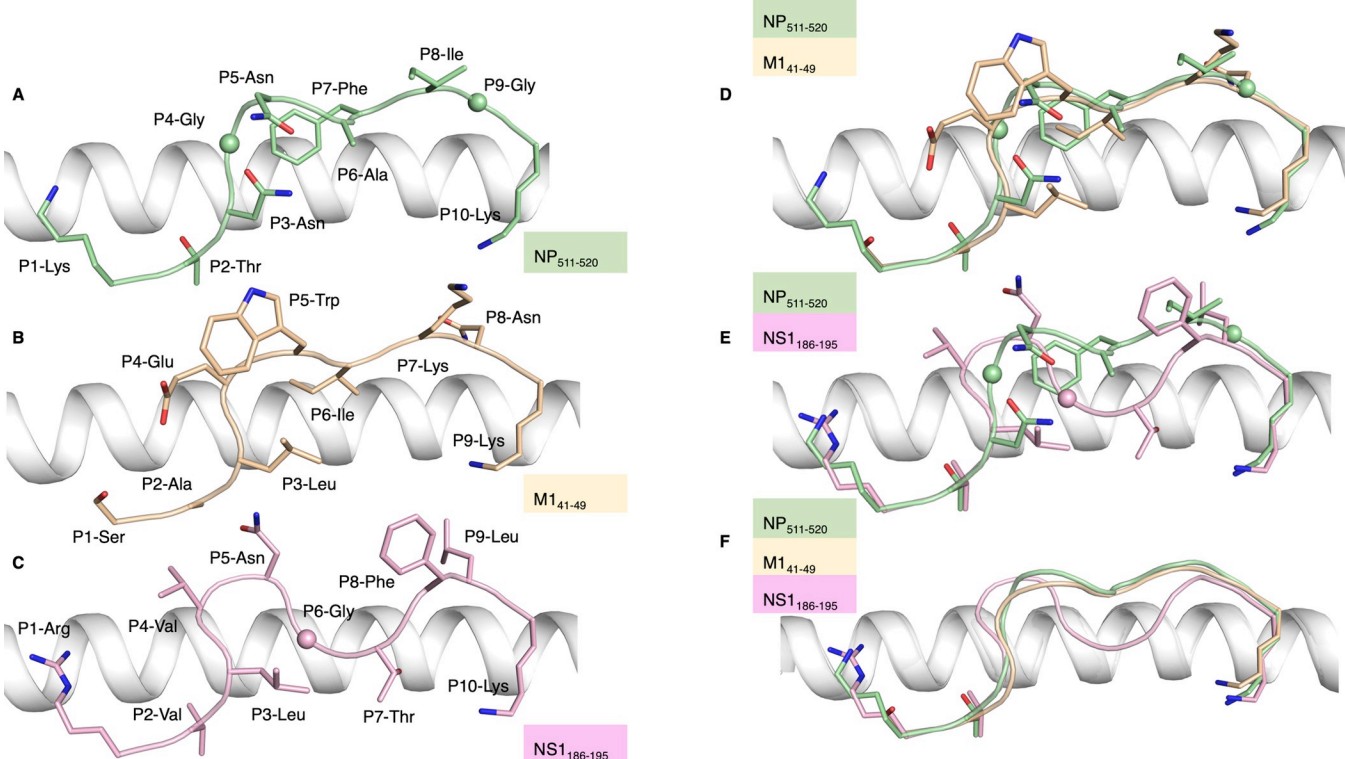

**Fig 6. Structures of IBV peptides complexed with HLA-A*11:01. (A-C)** NP$_{511-520}$ (green), M1$_{41-49}$ (sand) and NS1$_{186-195}$ (pink) peptides shown as sticks in complex with HLA-A*11:01 (grey). **(D-F)** Overlay of **(D)** NP$_{156-166}$ (green) and M1$_{41-49}$ (sand) and **(E)** NP$_{511-520}$ (green) and NS1$_{186-195}$ (pink), depicted as sticks, or **(F)** NP$_{511-520}$ (green), M1$_{41-49}$ (sand) and NS1$_{186-195}$ (pink) peptides with P1, P2 and PΩ depicted as sticks and remaining residues as cartoons, in complex with HLA-A*11:01 (grey).

displaying three residues with large side chains (P5-Asn, P7-Phe and P8-Ile), which might limit the number of potential T cell receptor (TCR) interactions. This flat conformation of the peptide is similar to the one observed for the HLA-A*02:01-restricted IAV M1$_{58-66}$ peptide [27,28] (S4 Fig).

The 9-mer M1$_{41-49}$ peptide also adopted a canonical conformation within the cleft of HLA-A*11:01 (Fig 6B). The M1$_{41-49}$ peptide contains anchor residues at P2-Ala and P9-Lys as well as secondary anchor residues P3-Leu and P6-Ile. Similar to the NP$_{511-520}$ peptide, the P9-Lys of the M1$_{41-49}$ is buried in the F pocket of HLA-A*11:01, forming a salt bridge with Asp77 instead of Asp116 (S3B Fig), and interacts with the hydrophobic patch formed by Ile95, Ile97 and Leu81 (S3C Fig). Although the backbone of the M1$_{41-49}$ peptide aligned with the one observed for NP$_{511-520}$ peptide (Fig 6D), there are four solvent exposed residues (P4-Glu, P5-Trp, P7-Lys and P8-Asn) in the M1$_{41-49}$ peptide, which therefore represent a larger surface accessible for TCR recognition. Structural overlay of the bound NP$_{511-520}$ and M1$_{41-49}$ peptides, with an average root mean square deviation (r.m.s.d.) of 1.52 Å, showed that despite sharing a similar backbone conformation, the overall peptide conformations are different due to the exposed side chains (Fig 6D). The antigen binding cleft of HLA-A*11:01 for both structures adopted a similar conformation with an average r.m.s.d. of 0.45 Å.

The 10-mer NS1$_{186-195}$ peptide anchors to HLA-A*11:01 by P2-Val and P10-Lys (Fig 6C) as well as secondary anchors P3-Leu and P7-Thr. P10-Lys is buried deep in the F pocket of the HLA-A*11:01 (S3D Fig). Overlay of the NS1$_{186-195}$ and NP$_{511-520}$ peptides display strikingly distinct conformations with an average r.m.s.d. of 1.7 Å for the Cα atoms, and only shared

homology for their main anchor residues (Fig 6E). Despite these striking differences, the antigen binding cleft of HLA-A*11:01 for both complexes showed little change with an average r. m.s.d. of 0.44 Å.

Overall, all three structures of HLA-A*11:01 presenting peptides derived from NP, M1 and NS1 IBV proteins provide an understanding into peptide repertoire specificity in the P2 and P9 pockets. They also show distinct peptide presentations (Fig 6F), which would enable the activation of CD8+ T cells with distinct TCR repertoires, providing broad CD8+ T cell coverage.

## Immunodominance of HLA-A*02:01- and HLA-A*11:01-restricted epitopes

HLA-A*02:01 is frequently expressed in Indigenous Australians, including the *LIFT* cohort [22] as well as in non-Indigenous people (Fig 1B), and has several IAV and IBV CD8+ T cell epitopes that have been well characterized [9,27]. To assess the immunodominance between HLA-A*02:01 and HLA-A*11:01-restricted IAV and IBV epitopes, PBMCs from HLA-A*02:01+/A*11:01+ donors were expanded against infected autologous PBMCs (A/X31 or B/Malaysia/2506/04) for 10 days before re-stimulating with C1R-A*11:01 cells pulsed with the immunogenic peptides identified in this study, as well C1R-A*02:01 cells pulsed with known HLA-A*02:01-restricted $M1_{58-66}$ and $PB1_{413-421}$ from IAV and $HA_{543-551}$, $NS1_{266-274}$ and $PB1_{413-421}$ from IBV [9] (Fig 7A). PBMCs were also re-stimulated with C1R-A*11:01 or -A*02:01 cells infected with IAV or IBV to determine total CD8+ T cell responses towards each virus. Following re-stimulation of IAV-stimulated PBMCs with IAV peptides, it was apparent that HLA-A*02:01-restricted $M1_{58-66}$ was immunodominant in comparison to HLA-A*11:01-restricted $PB1_{659-669}$ and $PB2_{320-331}$ (Fig 7B). In contrast to this, re-stimulation of IBV-stimulated PBMCs with IBV peptides displayed co-dominance between HLA-A*02:01 and -A*11:01, as no significant differences were observed (Fig 7C).

While the PBMCs were initially stimulated with virus-infected autologous PBMCs, they were restimulated by peptide-pulsed C1R cells transduced with either HLA-A*11:01 or -A*02:01 to isolate the presenting HLA allotype, confirming the observed CD8+ T cell responses observed to the seven immunogenic HLA-A*11:01-restricted epitopes are indeed restricted to HLA-A*11:01 in our assay.

HLA-A*11:01 restriction across all prior assays is supported by the highly dissimilar peptide binding preferences of HLA-A*02:01 and HLA-A*11:01. Whilst HLA-A*11:01 favours peptides possessing Val, Ser and Thr at P2 and basic (Arg and Lys) residues at the C-terminus (Fig 3B), HLA-A*02:01 favours those possessing Leu and Met at P2 and hydrophobic (Val and Leu) residues at the C-terminus [9]. As such the peptides anticipated to bind these two allotypes are highly distinct, with the majority of the peptides tested here (including the 7 immunogenic peptides) predicted to bind HLA-A*11:01 at lower nM concentrations than HLA-A*02:01, as calculated by NetMHC4.0 [29,30] (S5 Fig). Moreover, while it is possible that enzymatic processing of these peptides might generate improved binders for HLA-A*02:01 (as compared to the full-length sequences), assessment with NetMHC4.0 did not identify any shorter sequences within the immunogenic peptides that were predicted to bind HLA-A*02:01. We therefore have no evidence that the 7 HLA-A*11:01-restricted immunogenic peptides are presented by HLA-A*02:01 during infection, suggesting it is unlikely that the responses observed against these peptides are restricted to HLA-A*02:01.

## Immunogenic peptide conservation and predictability

To determine the sequence conservation of the immunogenic peptides identified, conservation analysis was performed. IAV and IBV protein sequences were sourced from the NCBI

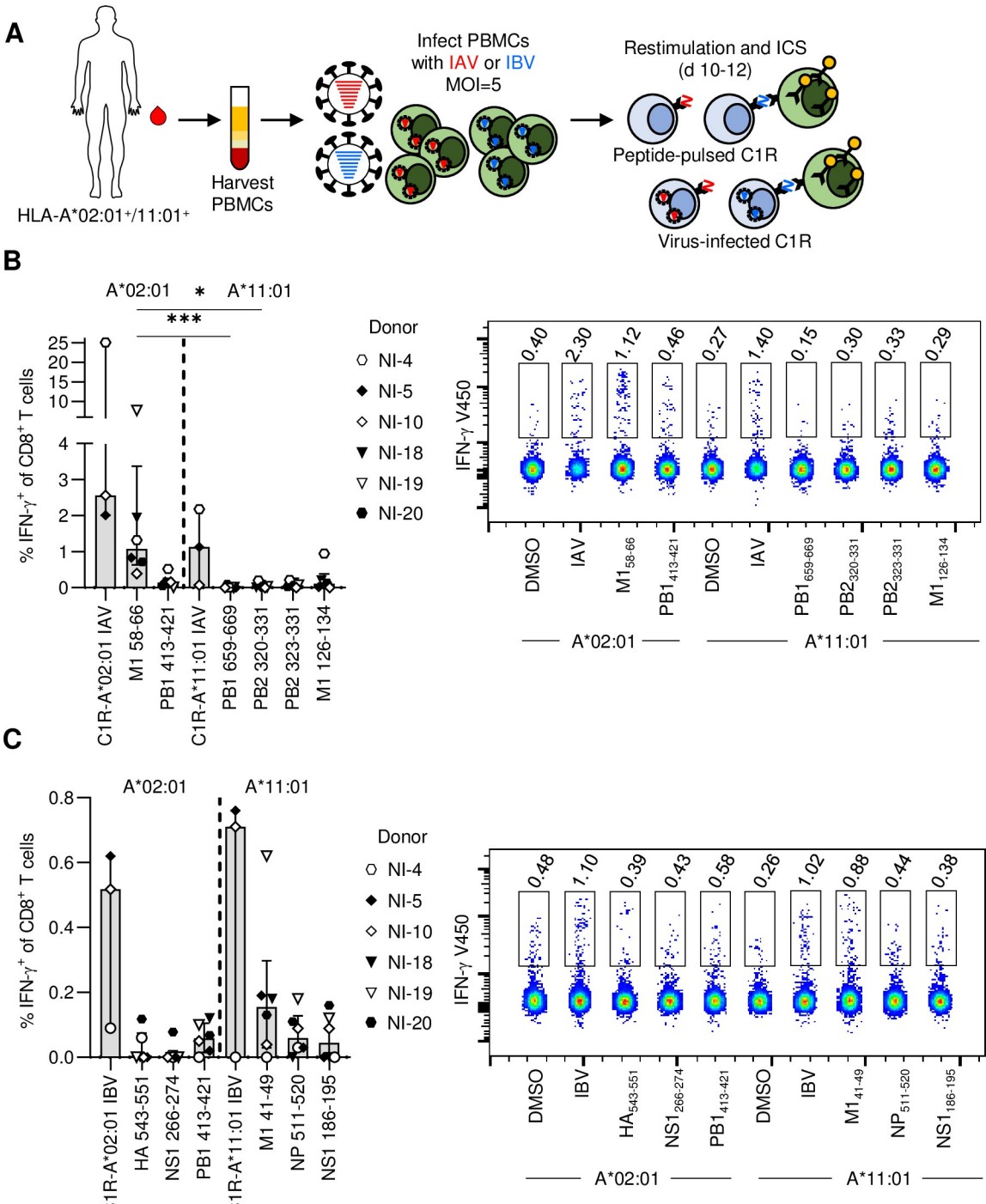

**Fig 7. Immunodominance of HLA-A\*02:01 and HLA-A\*11:01 IAV and IBV epitopes.** (**A**) PBMCs harvested from the peripheral whole blood of HLA-A\*02:01+/-A\*11:01+ donors were infected with either A/X31 or B/Malaysia/2506/04. PBMCs were expanded for 10 days and re-stimulated with C1R cells transduced with HLA-A\*02:01 or -A\*11:01 peptide-pulsed with HLA-A\*02:01 or -A\*11:01-restricted influenza peptides, respectively, to determine their immunodominance. (**B, C**) Frequency of IFN-γ+CD8+ T cells after re-stimulation with IAV- or IBV-infected C1R cells, or A\*02:01- or A\*11:01-restricted (**B**) IAV or (**C**) IBV peptides. Median and IQR are shown (n = 3, one experiment, n = 6, two independent experiments). (**B, C**) Representative concatenated FACS plots are shown for one donor depicting IFN-γ responses within CD8+ T cells for each stimulation condition. Friedman test with Dunn's multiple comparisons was used to determine significant differences between peptide stimulation conditions.

**Table 1. Conservation analysis of immunogenic influenza peptides.**

| Peptide | Sequence | Responders/Tested | Type/Subtype | Year | Conservation (%) |
|---|---|---|---|---|---|
| $M1_{126-134}$ | SCMGLIYNR | 2/4 | A/H1N1 | 1918–1957 | 99.3 |
| | | | A/H2N2 | 1957–1968 | 100 |
| | | | A/H3N2 | 1968–2019 | 100 |
| | | | A/H1N1 | 1977–2008 | 100 |
| | | | A/H1N1 | 2009–2019 | 99.9 |
| $PB1_{659-669}$ | AVATTHSWIPK | 2/7 | A/H1N1 | 1918–1957 | 96.4 |
| | | | A/H2N2 | 1957–1968 | 95.7 |
| | | | A/H3N2 | 1968–2019 | 99.8 |
| | | | A/H1N1 | 1977–2008 | 91.0 |
| | | | A/H1N1 | 2009–2019 | 99.6 |
| $PB2_{323-331}$ | FSFGGFTFK | 2/7 | A/H1N1 | 1918–1957 | 100 |
| | | | A/H2N2 | 1957–1968 | 100 |
| | | | A/H3N2 | 1968–2019 | 99.9 |
| | | | A/H1N1 | 1977–2008 | 99.9 |
| | | | A/H1N1 | 2009–2019 | 99.9 |
| $PB2_{320-331}$ | SSSFSFGGFTFK | 2/7 | A/H1N1 | 1918–1957 | 100 |
| | | | A/H2N2 | 1957–1968 | 100 |
| | | | A/H3N2 | 1968–2019 | 99.9 |
| | | | A/H1N1 | 1977–2008 | 99.9 |
| | | | A/H1N1 | 2009–2019 | 99.9 |
| $M1_{41-49}$ | SALEWIKNK | 7/8 | IBV | 1936–2019 | 99.9 |
| $NS1_{186-195}$ | RVLVNGTFLK | 4/8 | IBV | 1936–2019 | 99.9 |
| $NP_{511-520}$ | KTNGNAFIGK | 5/9 | IBV | 1936–2019 | 90.7 |

Influenza Virus Resource. Conservation analysis involved determining the frequency of variations in the immunogenic peptide sequences for human IAVs (A/H1N1 1918–1957, A/H2N2 1957–1968, A/H3N2 1969–2019, A/H1N1 1977–2008, A/H1N1 2009–2019) and IBVs identified between 1936–2019. IAV peptides ($M1_{126-134}$, $PB1_{659-669}$, $PB2_{320-331}$ and $PB2_{323-331}$) were highly conserved (>91%) across human-adapted subtypes that arose in the 101 years prior to 2019. The conservation of IBV peptides was 99.9% for $M1_{41-49}$ and $NS1_{186-195}$ and 90.7% for $NP_{511-520}$ across all IBVs from both the Yamagata and Victoria lineages detected from 1936–2019 (Table 1). The lower conservation observed in $NP_{511-520}$ is predominantly due to variations in position 3 residue 513N, which was more commonly observed in years prior to 2008, and from 2008 onwards, 513S became the dominant strain.

Peptide prediction algorithms have been used previously to identify immunogenic epitopes. Prediction algorithms factor in HLA-specific binding motifs and the predicted binding affinity of a peptide to the HLA of interest. Using NetMHCpan 4.0, we found that three of the five immunogenic peptides described in this study, or a variant thereof, were in the top five predicted epitopes from their respective influenza virus protein (Table 2). Interestingly, $M1_{125-134}$ represented a predicted peptide which ranked 3rd in the prediction of HLA-A*11:01-restricted IAV M1 protein epitopes, however our experimental data suggest that the minimal peptide presented by HLA-A*11:01 was $M1_{126-134}$.

Overall, high conservation of IAV- and IBV-derived CD8+ T cell epitopes identified in our study suggest that these viral peptides could potentially be targeted in a broadly cross-reactive T cell vaccine against IAV and IBV strains to provide coverage for both Indigenous and non-Indigenous people expressing HLA-A*11:01.

**Table 2. Comparison of experimentally identified and predicted HLA-A*11:01-restricted influenza epitopes.**

| Identified epitopes | | | | | Epitope prediction | | | |
|---|---|---|---|---|---|---|---|---|
| Protein | Start | Sequence | Length | Rank | Start | Sequence | Length | MS?A |
| M1 | 126 | SCMGLIYNR | 9 | 91 | 13 | SIIPSGPLK | 9 | Y |
| | | | | | 179 | MVLASTTAK | 9 | Y |
| | | | | | 125 | ASCMGLIYNR | 10 | Y |
| | | | | | 12 | LSIIPSGPLK | 10 | Y |
| | | | | | 178 | RMVLASTTAK | 10 | N |
| PB1 | 659 | AVATTHSWIPK | 11 | 24 | 338 | SIAPIMFSNK | 10 | N |
| | | | | | 488 | GTFEFTSFFY | 10 | N |
| | | | | | 661 | ATTHSWIPK | 9 | Y |
| | | | | | 523 | MSIGVTVIK | 9 | Y |
| | | | | | 544 | AQMALQLFIK | 10 | N |
| PB2 | 320 | SSSFSFGGFTFK | 12 | 5 | 105 | MTNTVHYPK | 9 | Y |
| | 323 | FSFGGFTFK | 9 | 3 | 321 | SSFSFGGFTFK | 11 | Y |
| | | | | | 323B | FSFGGFTFK | 9 | Y |
| | | | | | 322 | SFSFGGFTFK | 10 | N |
| | | | | | 320 | SSSFSFGGFTFK | 12 | Y |
| BM1 | 41 | SALEWIKNK | 9 | 32 | 236 | SSMGNSALVK | 10 | Y |
| | | | | | 62 | ASICFLKPK | 9 | N |
| | | | | | 145 | GTLCALCEK | 9 | N |
| | | | | | 179 | MVSAMNTAK | 9 | N |
| | | | | | 187 | KTMNGMGKGEDVQK | 14 | N |
| BNS1 | 186 | RVLVNGTFLK | 10 | 2 | 187 | VLVNGTFLK | 9 | N |
| | | | | | 186 | RVLVNGTFLK | 10 | Y |
| | | | | | 78 | KAIGVKMMK | 9 | N |
| | | | | | 107 | SSSNSNCTK | 9 | N |
| | | | | | 238 | RILNSLFER | 9 | N |
| BNP | 511 | KTNGNAFIGK | 10 | 6 | 226 | STFAGSTLPR | 10 | Y |
| | | | | | 332 | VVLPISIYAK | 10 | Y |
| | | | | | 36 | ATLAPPSNK | 9 | Y |
| | | | | | 502 | KMMNDSMAK | 9 | N |
| | | | | | 163 | TIYFSPIRVTFLK | 13 | N |

A The predicted epitope was (Y) or was not (N) found in the mass spectrometry data

B Red font indicates predicted epitopes which were also identified experimentally

## Discussion

The importance of CD8+ T cells in the clearance of influenza virus infections is well documented in mice [31–33] and humans [34–36]. HLA-I molecules on antigen presenting cells prime epitope-specific CD8+ T cells to mount an adaptive immune response which aids in viral clearance and forms long-lived immunological memory. Given that CD8+ T cell recognition is determined by a spectrum of HLAs expressed in an individual, and that HLAs differ between ethnic groups, broadly protective cross-reactive CD8+ T cell immunity needs to be investigated specifically for HLAs that are dominant across different ethnicities. To date, CD8+ T cell epitopes for both IAVs and IBVs have been identified only for two class I HLAs, HLA-A*02:01 [9] and HLA-A*24:02 [23]. As shown in our study, HLA-A*11:01 is one of the most prevalent HLAs, especially in East Asia and Oceanian populations, with high enrichment

in Papa New Guinea Madang people (63.6%), China Yunnan Hani (61.3%), Taiwan Hakka (40.0%), Pakistan Brahui (25.2%), Vietnam Hanoi Kinh (22.9%), Cape York Peninsula Aboriginal people (18.0%) and New Zealand Maori (16.7%). As Indigenous populations globally are highly susceptible to influenza-induced morbidity and mortality [11–14], especially when a new influenza virus emerges, it is of critical importance to identify CD8+ T cell epitopes for prominent HLAs in Indigenous people to understand how to effectively protect these populations.

Here, HLA-A*11:01-specific CD8+ T cell responses were dissected to further understand immunity to IAV and IBV infections. Previous reports described a handful of HLA-A*11:01-restricted IAV epitopes [15–21], although most epitope-specific CD8+ T cell responses were detected in low frequencies or used a low number of donors. When these reported epitopes were tested using our CD8+ T cell expansion and ICS methods, the majority of published HLA-A*11:01-restricted peptides did not induce detectable IFN-γ by CD8+ T cells. This suggests that the HLA-A*11:01-restricted epitopes reported previously were either at low frequencies in our donors, or CD8+ T cells elicited towards those epitopes are found only in a select number of individuals, perhaps dependent on their TCR repertoires. The precursor frequencies of epitope-specific CD8+ T cells determines the magnitude of the response towards each of the peptides screened. The method used to identify the HLA-A*11:01-restricted epitopes could affect the interpretation of CD8+ T cell responses. The lack of IFN-γ response towards published IAV epitopes was unexpected and emphasized the need to identify HLA-A*11:01-restricted influenza virus epitopes that are presented by HLA-A*11:01 during infection and activate an influenza virus-specific CD8+ T cell response.

Using immunopeptidomics, we have embarked on a large epitope discovery study to define the CD8+ T cell landscape for HLA-A*11:01-expressing Indigenous and non-Indigenous people. Screening of 79 influenza A virus (IAV)- and 57 influenza B virus (IBV)-derived peptides naturally presented by HLA-A*11:01 during infection, revealed two main IAV epitopes (A11/PB2$_{320-331}$ and A11/PB2$_{323-331}$). We have also identified the first HLA-A*11:01-restricted IBV epitopes (A11/M$_{41-49}$, A11/NS1$_{186-195}$ and A11/NP$_{511-520}$). The number of immunogenic peptides identified from the pool generated by immunopeptidomics for HLA-A*11:01 closely resembles our previous studies for HLA-A*02:01 and HLA-A*24:02 [9,23].

The crystal structures generated provide a basis for the preference of anchor residues that sit within the B and F pockets of HLA-A*11:01 [37] observed by mass spectrometry. The B pocket of HLA-A*11:01 contains Tyr7, Tyr9, Tyr159 and Met45, that have large side chains making the B pocket relatively shallow optimal to bind small peptide residues such as Thr, Val, and Ser as shown by the mass spectrometry data. Meanwhile, the F pocket of HLA-A*11:01 contains small buried hydrophobic residues (Ile95, Ile97 and Leu81) as well as negatively charged Asp77 and Asp116 that form a salt bridge with the preferred Lysine at the C-terminal part of the peptide as shown by mass spectrometry. Therefore, the crystal structures are characteristic of the HLA-A*11:01 B and F pockets and give a rational for the preferred residues observed for 9-, 10- and 11-mer peptides eluted from HLA-A*11:01 molecule. These structures represent the first insight into IBV epitope presentation by HLA-A*11:01 molecule, and their diverse conformations suggest that a diverse T cell repertoire might be able to recognise them.

Identification of novel T cell epitopes for IBV is of particular importance as immunity to IBVs is greatly understudied. As there are no established animal reservoirs for IBVs, there is no potential for zoonotic transmission to humans, hence research has largely focused on IAV, leaving IBV understudied [3,38]. However, IBVs still have substantial clinical importance, accounting for ~25% of influenza cases annually [39, 40], causing mild to severe disease especially in 5–17 year-old children. IBVs can also lead to severe neurological, cardiovascular and muscular complications [41] and secondary bacterial pneumonia [41,42]. Because the effectiveness of the current vaccines against IBV is modest at ~ 50% [43], a universal T cell-based

IBV vaccine that provides broadly cross-reactive and long-lasting protection is of a great interest. In our study, we identified three prominent IBV CD8+ T cell epitopes restricted by the prevalent HLA-A*11:01, thus extending the epitope knowledge beyond already reported immunodominant IBV epitopes for HLA-A*02:01 [9] and HLA-A*24:02 [23].

Importantly, all the IAV- and IBV-derived peptides constituting IAV and IBV epitopes were >90% conserved among respective influenza viruses, indicating their suitability as vaccine targets to provide broad cross-reactive immunity across IAVs and IBVs.

The lack of variation in these peptide regions indicate their functional importance and that mutation might be detrimental to viral fitness. With conservation at >91% for each IAV and IBV peptide over the course of the past 100 years, the epitopes described here make ideal candidates for a CD8+ T cell peptide-based influenza virus vaccine. High conservation of the peptide sequence will allow for CD8+ T cells to provide long-lasting immunity. Once an epitope-specific CD8+ T cell memory pool is established, subsequent infections will be less severe and resolve faster than without pre-existing CD8+ T cell immunity. Overall, our findings provide insight for the development of rationally designed, broadly cross-reactive, influenza vaccines to protect HLA-A*11:01-expressing individuals from severe influenza disease, and have the potential to contribute to a universal influenza vaccine providing global coverage for prominent HLA types across different ethnicities.

## Materials and methods

### Ethics statement

Experimental work involving the use of human blood was conducted in line with Declaration of Helsinki Principles and according to the Australian National Health and Medical Research Council Code of Practice. Human blood samples were collected after obtaining signed informed consent from all participants. PBMCs were obtained from buffy packs (Australian Red Cross Lifeblood, West Melbourne, Australia) or whole blood from consenting donors. LIFT cohort participant PBMCs were obtained in collaboration with the Menzies School of Health Research (Charles Darwin University, NT, Australia), as previously described [22]. Human experimental work was approved by the University of Melbourne Human Ethics Committee (ID 1955465, 1443389), the Australian Red Cross Lifeblood Ethics Committee (ID 2015#8) and HREC of Northern Territory Department of Health and Menzies School of Health Research (ID 2012–1928). Human PBMCs were isolated and cryopreserved in liquid nitrogen until later use.

### Viruses, cell lines and peptides

A/X31 H3N2 and B/Malaysia/2506/04 influenza viruses were used in this study. Stable HLA-A*11:01 or HLA-A*02:01 class-I-reduced transductants (C1R-A*11:01 or C1R-A*02:01, respectively) used for virus or peptide stimulation of PBMCs were maintained in complete RPMI medium (RPMI-1640, 10% heat-inactivated fetal calf serum, 5 mM HEPES, 2 mM L-glutamine, 1 mM MEM sodium pyruvate, 100 μM non-essential amino acids, 55 μM 2-mercaptoethanol, 100 μg/mL streptomycin, 100 U/mL penicillin; Gibco, Thermofisher Scientific, Scoresby, VIC, Australia), or complete RPMI medium supplemented with 200μg/mL hygromycin B, respectively, and incubated at 37°C, 5% $CO_2$. Synthetic peptides were purchased from GenScript and reconstituted to 1mM in DMSO.

### Large-scale infection for immunopeptidome analysis

C1R-A*11:01 transductants were infected with IAV (A/X31) or IBV (B/Malaysia/2506/04) at a MOI of 5, harvested and snap-frozen as pellets of 9-12x10$^8$ cells 12 hrs post-infection, and

stored at -80˚C until use, as described previously [9,23]. HLA surface staining was performed with ~$10^6$ cells using anti-HLA class I PE-Cy7 (Biolegend), followed by fixation in 1% paraformaldehyde (ProSciTech). Infection was confirmed by fixation of cells in 1% paraformaldehyde, prior to intracellular staining with anti-NP FITC for IAV (Clone 1331, GeneTex Cat# GTX36902) or IBV (Clone H89B, ThermoFisher Cat# MA1- 7306) (1:200 in 0.3% saponin [Sigma] in PBS, 45min, 4˚C). As C1R-A*11:01 are GFP+, cells were then washed in PBS, and incubated with anti-mouse PE (Goat F(ab')2 Anti-Mouse IgG(H+L), Human ads-PE, Southern Biotech, cat# 1032–09) (1:150 in 0.3% saponin in PBS, 45min, 4˚C). For all stains, cells were washed in PBS and acquired by flow cytometry using a BD LSRII flow cytometer running BD FACSDiva software, then analyzed using FlowJo version 10 (BD).

## Liquid Chromatography-tandem mass spectrometry (LC-MS/MS) analysis of HLA-bound peptides

C1R-A*11:01 cells were lysed by cryomilling and detergent-based lysis in buffer consisting of 0.5% IGEPAL CA-630, 50 mM Tris-HCl pH8.0, 150 mM NaCl and protease inhibitors (cOmplete Protease Inhibitor Cocktail Tablet; Roche Molecular Biochemicals), and the HLA class I immunoaffinity purified using the pan class I antibody W6/32 as described previously [24]. Peptide/MHC complexes were dissociated (10% acetic acid) and fractionated by reversed phase high performance liquid chromatography (RP-HPLC), collecting 500μL fractions throughout the gradient as described [44]. 9 pools of peptide containing fractions were generated, vacuum-concentrated, and reconstituted in 15μL 0.1% formic acid (Honeywell) in Optima LC-MS water, containing 0.25pmol iRT internal standard peptides [45]. Fraction pools were analyzed by LC-MS/MS on a Q-Exactive Plus Hybrid Quadrupole Orbitrap (Thermo Fisher Scientific) coupled to a Dionex UltiMate 3000 RSLCnano system (Thermo Fisher Scientific). 6 μl injections were loaded onto an Acclaim PepMap 100 Trap column (100 μm x 2 cm, nanoViper, C18, 5 μm, 100Å; Thermo Scientific) in 2% acetonitrile, 0.1% formic acid at 15 μl/min. Peptides were eluted over an Acclaim PepMap RSLC Analytical column (75 μm x 50 cm, nanoViper, C18, 2 μm, 100Å; Thermo Scientific) with an increasing gradient of buffer B (80% acetonitrile, 0.1% formic acid) of 2.5–7.5% over 1 min, 7.5–32.5% over 55 min, 32.5–40% over 5 min, 40–99% over 5 min, 99% over 6 min and returning to 2.5% buffer B over 1 min, before re-equilibration at 2.5% for 20 min at a flow rate of 250 nL/min. Data were collected in positive mode: MS1 resolution, 70,000; scan range, 375–1,600 m/z; MS2 resolution, 35,000; dynamic exclusion, 15 s. The top 12 ions of +2 to +6 charge per cycle were subject to MS/MS.

## Bioinformatic analysis of mass spectrometry data

For peptide assignments, C1R-A*11:01 LC-MS/MS data sets were searched as a single project using PEAKS 8.5 (Bioinformatics Solutions Inc.) database search. Data were searched against the reviewed human proteome (downloaded from Uniprot April 2016) with the proteome and 6 frame translated nucleotide sequence of either A/X31 or B/Malaysia/2506/04 appended, incorporating a contaminant database of common LC-MS/MS contaminants, and performing False Discovery Rate (FDR) analysis by decoy-fusion. For initial searches the following settings were employed: Instrument–Orbitrap (Orbi-Orbi), Fragment–HCD, Parent Mass Error Tolerance—10.0 ppm, Fragment Mass Error Tolerance—0.02 Da, Precursor Mass Search Type—monoisotopic, Enzyme—None, Variable Modifications—Oxidation (M) 15.99 and Deamidation (NQ) 0.98, Max Variable PTM Per Peptide—3. Data from previous analyses of presentation by endogenous HLA of C1R cells (endogenous HLA data sets: A/X31 [23], B/Malaysia/ 2506/04 [9]) were searched via the same pipeline. A second search of C1R-A*11:01 LC-MS/MS

data against the human + A/X31 database was performed incorporating additional variable modifications: Oxidation (M) 15.99, Deamidation (NQ) 0.98, Dioxidation (M) 31.99, Dihydroxy (YWFRKPC) 31.99, Cysteine oxidation to cysteic acid (C) 47.98.

For motif analysis, a peptide spectrum match (PSM) FDR threshold of 1% was set. The HLA-A*11:01 length distribution and binding motif was generated based on assignments to the human proteome in the human + B/Malaysia/2506/04 database search, filtered of peptides identified in endogenous HLA data sets (both class I and II) or within the contaminant database alone. Redundancy based on ambiguity of Ile/Leu assignment was minimal (<1%) and both sequence assignments were maintained within the analysis. Sequence Logos were generated with Seq2Logo 2.0 using default settings [46], graphs were generated using GraphPad Prism 8.0.2 for Windows (GraphPad Software, San Diego, California USA, www.graphpad.com). To identify potential influenza-derived HLA ligands, assignments to the influenza proteome/translation were considered for PEAKS peptide scores (-10lgp) both above and below the 1% FDR threshold. Peptide assignments (length <30 amino acid) mapping to the influenza proteome or 6 frame genome translation were considered valid if detected in the infected samples but not the uninfected samples. If a peptide assignment mapping to influenza was found in both infected and uninfected samples, spectra and retention times were compared. If spectra/retention times were distinct, and the higher scoring assignment was within the infected samples, identifications were maintained within the analysis. Assignments included within the analysis are contained in S1 Dataset.

Predicted binding affinities of stripped sequences of 8-14-mers to HLA-A*11:01, -B*35:03 and -C*04:01 were calculated using NetMHC4.0 [29,30], and Strong binders (SB) and Weak Binders (WB) assigned based on the default cut-offs of % Rank (0.5 and 2, respectively). Potential binders (PB) were assigned based on evidence of pull down, overlap with predicted binders, and observation in endogenous HLA data sets, including class II peptides which can contaminate isolations.

The mass spectrometry HLA-A*11:01 immunopeptidome data sets have been deposited to the ProteomeXchange Consortium via the PRIDE [47] partner repository with the dataset identifier PXD028985 and 10.6019/PXD028985.

## Infection of C1R cells with influenza viruses

C1R-A*11:01 or C1R-A*02:01 cells were infected with IAV (A/X31) or IBV (B/Malaysia/2506/2004) for use in CD8+ T cell stimulations. C1Rs were infected using a MOI of 5 in RPMI medium for 1 h at 37˚C/5% $CO_2$ before adding complete RPMI and incubating a further 11 h. Infected C1Rs were washed twice before use for CD8+ T cell stimulation.

## Expansion of virus- or antigen-specific CD8+ T cells

PBMCs were removed from storage in liquid nitrogen, thawed and washed twice in RPMI. Virus-specific expansions involved mixing PBMCs with IAV- or IBV-infected C1R cells at a 10:1 ratio. Antigen-specific expansions were performed as previously described [23] by peptide-pulsing one-third of PBMCs with pooled or individual influenza virus peptides for 1 h before washing and mixing with non-stimulated PBMCs. CD8+ T cell cultures were maintained in complete RPMI and incubated at 37˚C/5% $CO_2$ for 4 days before adding and maintaining a concentration of 20 U/mL of recombinant human IL-2. Virus-specific expansions were re-stimulated with influenza virus-infected C1Rs at a 1:10 stimulator-to-responder ratio on day 8 and maintained as described above.

## CD8⁺ T cell re-stimulation and intracellular cytokine staining

Intracellular cytokine staining (ICS) was performed on day 10–12 to identify epitope-specific CD8⁺ T cells after influenza virus or peptide stimulation. Virus-specific CD8⁺ T cells were re-stimulated with peptide-pulsed C1R-A*11:01 cells at a 1:2 ratio and antigen-specific CD8⁺ T cells were re-stimulated with 1 μM of individual or pooled influenza peptides. Re-stimulations were done for 5 h in the presence of brefeldin A (GolgiPlug, BD Biosciences), monensin (GolgiStop, BD Biosciences) and anti-CD107a-AF488 antibody (Invitrogen). After stimulation cells were surface stained with LiveDead NIR fixable viability dye (Invitrogen), anti-CD3-PE-Cy7, anti-CD4-PE and anti-CD8-PerCP-Cy5.5 (BD Biosciences) before permeabilizing for intracellular staining with anti-IFN-γ-V450, anti-TNF-AF700 and anti-MIP1-β-APC antibodies (BD Biosciences). The gating strategy for determining CD8⁺ T cell activation is shown in S6 Fig.

## Peptide conservation analysis

Protein sequences for IAVs or IBVs were sourced from the NCBI influenza virus resource (https://www.ncbi.nlm.nih.gov/genomes/FLU/Database/nph-select.cgi?go=database). IAV sequences from the dominant circulating strain for the relevant years were used to assess conservation, while all IBV sequences from 1936–2019 were analyzed together. BioEdit (version 7.2.5) [48] was used to remove incomplete sequences from the dataset. Conservation of each amino acid residue of the relevant peptide sequences were determined using Unipro UGENE software [49]. The conservation frequencies of each residue were averaged to determine the conservation of the entire peptide sequence.

## Immunodominance/PBMC infection

PBMCs were removed from storage in liquid nitrogen, thawed and washed twice in RPMI. One-tenth of PBMCs were infected with either IAV or IBV, while the remaining cells were added to the tissue culture plate and incubated at 37˚C/5% $CO_2$. PBMCs for infection were mixed with IAV (A/X31) or IBV (B/Malaysia/2506/04) at a MOI of 5 in serum-free RPMI and incubated at 37˚C/5% $CO_2$ for 1 h before fetal-calf serum was added and incubated for a further 3 h. Infected PBMCs were washed with RPMI to remove any remaining virus and mixed with the uninfected PBMCs in the tissue-culture plate. PBMCs were maintained in complete RPMI medium and incubated at 37˚C/5% $CO_2$ for 4 d before adding and maintaining a concentration of 20 U/mL of recombinant human IL-2 (Roche). On day 10 PBMCs were re-stimulated with IAV- or IBV-infected C1R-A*02:01 or -A*11:01 cells (as described above), or HLA-A*02:01- or HLA-A*11:01-restricted peptides derived from either virus and ICS was performed.

## Protein expression, purification and crystallisation

Soluble HLA-A*11:01 heterotrimers containing either $NP_{511-520}$, $M1_{41-49}$, or $NS1_{186-195}$ peptide, were prepared as previously described [50]. In summary, a truncated HLA-A*11:01 construct containing the extracellular region of the HLA molecule (residues 1–276), and human beta-microglobulin (β2m) were expressed separately in a BL21-RIL *Escherichia coli* strain as inclusion bodies. The inclusion bodies were subsequently extracted, washed and resuspended into a solution containing 6 M guanidine. Each pHLA complex was then refolded into a cold refolding solution (3 M Urea, 100 mM Tris-HCl pH 8, 2 mM Na-EDTA, 400 mM L-arginine-HCl, 0.5 mM oxidized glutathione, 5 mM reduced glutathione) by adding 30mg of HLA heavy chain, 20mg of β2m and 4mg of peptide. The refolding solution was then dialysed in 10mM

Tris-HCl pH 8, and the protein was purified by a succession of affinity column chromatography.

## Crystallisation, data collection and structure determination

Crystals of the pHLA-A*11:01 complexes were grown by the hanging-drop, vapour-diffusion method at 20°C with a protein/reservoir drop ratio of 1:1 with seeding at a concentration of 6 mg/mL in the following conditions: $M1_{41-49}$: 2M Ammonium Sulfate, 0.1M Tris-HCl pH 8, 0.2M Lithium Sulfate, 2% PEG400, $NP_{511-520}$: 20% PEG 3350, 0.1M BisTris pH 6.5 and $NS1_{186-195}$: 26% 3350, 0.1M BisTris pH 6.3, 0.2M Lithium Sulfate. The crystals were soaked in a cryo-protectant solution containing mother liquor solution with the PEG concentration increased to 30% (w/v) and then flash frozen in liquid nitrogen. The data were collected on the MX1 and MX2 beamlines [51]. The data was processed using XDS [52] and the structures were determined by molecular replacement using the *Phaser* program [53] from the CCP4 suite (1994) with a model of HLA-A*11:01 without the peptide (derived from PDB ID: 4MJ5 [21]. We observed for HLA-A*11:01-$M1_{41-49}$ complex that the diffraction was anisotropic, as a result the density map quality was poor. To improve the quality of the initial map, we used the Diffraction Anisotropy Server from UCLA [54] that apply ellipsoidal truncation and anisotropic scaling to the diffraction data. As per our observation, the analysis showed that the data has strong anisotropy with a score of 30.1 out of 100. Manual model building was conducted using the Coot software [55] followed by maximum-likelihood refinement with the Buster program [56]. The final model has been validated using the Protein Data Base validation web site and the final refinement statistics are summarized in S6 Table. All molecular graphics representations were created using PyMol [57]. The final crystal structure models for the HLA-A*11:01 complexes have been deposited to the Protein DataBank (PDB) under the following accession codes: HLA-A*11:01- $NP_{511-520}$: 7S8Q, HLA-A*11:01- $M1_{41-49}$: 7S8R and HLA-A*11:01-$NS1_{186-195}$: 7S8S.

## Thermal stability assay

Thermal shift assays were performed to determine the stability of each pHLA-A*11:01 complex using fluorescent dye, SYPRO Orange, to monitor protein unfolding. The thermal stability assay was performed in the Real Time Detection system (Corbett RotorGene 3000), originally designed for PCR. Each pHLA complex was measured in 10 mM Tris-HCl pH8, 150 mM NaCl, at two concentrations (5 and 10 μM) in duplicate (n = 1), and was heated from 25 to 95°C with a heating rate of 1°C/min. The fluorescence intensity was measured with excitation at 530 nm and Emission at 555 nm. The Tm, or thermal midpoint, represents the temperature for which 50% of the protein is unfolded. The results are reported in the S5 Table.

## Supporting information

**S1 Fig. Locations of discussed alternative reading frame peptides.** Translation of the 5' end of A/X31 segment 7 (M) mRNA in three translation frames. The N-terminal portion of M1 is shown in blue with the region also contributing to the N-terminal ectodomain of M2 outlined in red (M2 segment). The alternative ectodomain (M42 segment), encoded in frame 3, is outlined in purple encompassing both an alternative start site and a detected HLA-A*11:01 ligand ($M(+3)_{39-48}$, light green). A second HLA-A*11:01 ligand encoded in frame 3 ($M(+3)_{94-103}$, light green), preceded by another potential alternative start site, is also shown. Numbering below relates to translation in frame 3, starting at the viral 5' UTR. Red asterisks denote stop codons and dark green M are known/proposed start sites.
(TIF)

**S2 Fig. Electron density maps for each HLA-A\*11:01-IBV complexes.** Density map for the structures of the three HLA-A\*11:01 (white cartoon) binding to IBV peptides coloured in light green ($NP_{156-166}$), sand ($M1_{41-49}$) and light pink ($NS1_{186-195}$). The omit 2Fo-Fc map is contoured at 3 sigma (**A-C**) or 2.5 sigma (**B**) and coloured in green on the top panels, while the electron density after refinement is shown by a blue Fo-Fc map contoured at 1 sigma on the bottom panels (**D-F**).
(TIF)

**S3 Fig. Structures of the PΩ of the IBV peptides interacting with the cleft of HLA-A\*11:01.** (**A**) The binding cleft of HLA-A\*11:01 (white cartoon) with the peptide $NP_{156-166}$ (light green stick) interacting with residues within the binding cleft (white stick). (**B**) The binding cleft of HLA-A\*11:01 (white cartoon) with the peptide $M1_{41-49}$ (sand stick) interacting with residues within the binding cleft (white stick). (**C**) Overlay of HLA-A\*11:01 (white cartoon) presenting $NP_{156-166}$ peptide (light green stick) and HLA-A\*11:01 (white cartoon) presenting $M1_{41-49}$ peptide (sand stick) interacting with residues within the binding cleft (white stick). (**D**) The binding cleft of HLA-A\*11:01 (white cartoon) with the peptide $NS1_{186-195}$ (pink stick) interacting with residues within the binding cleft (white stick).
(TIF)

**S4 Fig. Comparison between HLA-A\*11:01-$NP_{156-166}$ and HLA-A\*01:01-$M1_{58-66}$.** Overlay of HLA-A\*11:01 (white cartoon) presenting $NP_{156-166}$ peptide (light green stick) and HLA-A\*02:01 (orange cartoon) presenting $M1_{58-66}$ (orange stick).
(TIF)

**S5 Fig. Predicted binding of HLA-A\*11:01-restricted peptides.** (**A-B**) Predicted affinity for HLA-A\*11:01-restricted (**A**) IAV-derived and (**B**) IBV-derived peptides for HLA-A\*02:01, A\*11:01, B\*35:03 and C\*04:01, calculated using NetMHC4.0. Each line represents an individual peptide, with immunogenic peptides highlighted by colored lines as shown in the legends.
(TIF)

**S6 Fig. Determining CD8+ T cell activation by flow cytometry.** Gating strategy used to determine CD8+ T cell activation and polyfunctionality defined by IFN-γ, TNF, MIP1-β and/ or CD107a expression.
(TIF)

**S1 Dataset. Influenza-derived peptides identified by mass spectrometry.** Influenza-derived peptides identified across the 7 C1R-A\*11:01 data sets by different search strategies. Each sheet represents a different search strategy as described in the descriptor sheet.
(XLSX)

**S1 Table. Demographics of Indigenous and non-Indigenous donors.**
(DOCX)

**S2 Table. Previously reported HLA-A\*11:01-restricted peptides from influenza A virus.**
(DOCX)

**S3 Table. HLA-A\*11:01-restricted IAV peptides identified by mass spectrometry.**
(DOCX)

**S4 Table. HLA-A\*11:01-restricted IBV peptides identified by mass spectrometry.**
(DOCX)

**S5 Table. Thermal stability of peptide-HLA-A*11:01 complexes.**
(DOCX)

**S6 Table. Data collection and refinement statistics.**
(DOCX)

**S1 Source Data.**
(XLSX)

## Acknowledgments

Computational resources for proteomics analysis were supported by the R@CMon/Monash Node of the NeCTAR Research Cloud, an initiative of the Australian Government's Super Science Scheme and the Education Investment Fund. We thank the Monash Macromolecular Crystallization Facility staff, and the staff at the Australian synchrotron for technical assistance. This research was undertaken in part using the MX2 beamline at the Australian Synchrotron, part of ANSTO, and made use of the Australian Cancer Research Foundation (ACRF) detector. We acknowledge the Melbourne Cytometry Platform at the Peter Doherty Institute for Infection and Immunity for provision of flow cytometry services, and the Monash University Technology Research platforms, the Monash Proteomics & Metabolomics Facility and Flow-Core, for the provision of instrumentation, training, and technical support.

## Author Contributions

**Conceptualization:** E. Bridie Clemens, Weisan Chen, Jamie Rossjohn, Stephanie Gras, Anthony W. Purcell, Luca Hensen, Katherine Kedzierska, Patricia T. Illing.

**Data curation:** Steven Y. C. Tong.

**Formal analysis:** Jennifer R. Habel, Andrea T. Nguyen, Louise C. Rowntree, Christopher Szeto, Liyen Loh, Stephanie Gras, Luca Hensen, Patricia T. Illing.

**Funding acquisition:** Stephanie Gras, Anthony W. Purcell, Katherine Kedzierska.

**Investigation:** Jennifer R. Habel, Andrea T. Nguyen, Louise C. Rowntree, Christopher Szeto, Liyen Loh, Stephanie Gras, Luca Hensen, Patricia T. Illing.

**Resources:** Weisan Chen, Steve Rockman, Jane Nelson, Jane Davies, Adrian Miller.

**Supervision:** Stephanie Gras, Anthony W. Purcell, Luca Hensen, Katherine Kedzierska.

**Writing – original draft:** Jennifer R. Habel, Luca Hensen, Katherine Kedzierska, Patricia T. Illing.

**Writing – review & editing:** Jennifer R. Habel, Andrea T. Nguyen, Louise C. Rowntree, Christopher Szeto, Nicole A. Mifsud, E. Bridie Clemens, Liyen Loh, Weisan Chen, Steve Rockman, Jane Nelson, Jane Davies, Adrian Miller, Steven Y. C. Tong, Jamie Rossjohn, Stephanie Gras, Anthony W. Purcell, Luca Hensen, Katherine Kedzierska, Patricia T. Illing.

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
