## [Decision Letter · Decision Letter 0]

23 Nov 2021

Dear A/Prof Kedzierska,

Thank you very much for submitting your manuscript "HLA-A*11:01-restricted CD8+ T cell immunity against influenza A and influenza B viruses in Indigenous and non-Indigenous people" for consideration at PLOS Pathogens. As with all papers reviewed by the journal, your manuscript was reviewed by members of the editorial board and by several independent reviewers. In light of the reviews (below this email), we would like to invite the resubmission of a significantly-revised version that takes into account the reviewers' comments.

We cannot make any decision about publication until we have seen the revised manuscript and your response to the reviewers' comments. Your revised manuscript is also likely to be sent to reviewers for further evaluation.

Sincerely,

Christian Munz

Associate Editor

PLOS Pathogens

Sonja Best

Section Editor

PLOS Pathogens

Kasturi Haldar

Editor-in-Chief

PLOS Pathogens

orcid.org/0000-0001-5065-158X

Michael Malim

Editor-in-Chief

PLOS Pathogens

orcid.org/0000-0002-7699-2064

Reviewer's Responses to Questions

**Part I - Summary**

Reviewer #1: In this manuscript, Habel et al. aim at characterizing HLA-A*11:01-restricted CD8+ T cell responses in HLA-A*11:01-expressing Indigenous and non-Indigenous Australian donors. To this end, the authors used an experimental approach, which combined mass spectrometry analysis to identify peptides expressed by HLA-A*11:01 and in vitro T cell stimulation for testing their immunogenicity. Although the results of this analysis may help increase our knowledge of the CD8+ T cell response against Influenza viruses, data provided in the study are limited (especially for IAV reactivity) and do not allow to make general conclusion on immunodominant epitopes involved in potentially protective CD8+ immune response in HLA-A*11:01+ individuals. Overall, developing a universal influenza vaccine based on this evidence would be quite ambitious.

Reviewer #2: This is an interesting and well-performed study of HLA-A*11:01-restricted CD8+ T cell responses to influenza A and B virus peptides. HLA-A*11:01 is an allele prevalent in East Asia and Oceania, and highly expressed in indigenous populations at high risk of influenza infection. Current influenza vaccines are designed primarily to induce neutralizing antibodies against hemagglutinin and neuraminidase. Improving the efficacy of influenza A and B vaccines may be achieved by targeting and activating influenza-specific cytotoxic CD8+ T cells and memory formation.

The authors used Masspec to identify influenza A- or B-derived peptides eluted from HLA class I after infection of APCs in vitro. 79 influenza A and 57 influenza B derived peptides were subsequently screened for immunogenicity in HLA-A*11:01-positive PBMC assays. The authors identified three immunodominant influenza A- and three influenza B-derived peptides from PB1, PB2, M, NS1 and NP proteins that appeared highly conserved among influenza viruses.

The study also includes data on structures of HLA-A*11:01 in complex with influenza B-derived peptides identified. Since this lies outside of my field of expertise, I will not comment on this part of the study.

This study's results may inform the design of future influenza vaccines more suitable to meet the needs of indigenous populations, but also the design of more effective influenza vaccines in general, by focusing on viral proteins distinct from hemagglutinin or neuraminidase, targeted by CD8+ T cells.

**Part II – Major Issues: Key Experiments Required for Acceptance**

Reviewer #1: Given the variability in the T cell responses and the low number of donors analyzed, this reviewer believes that including more donors in the screenings shown in Fig. 2 and 4 may help for a better definition of statistical significance in immunodominant epitopes presented on HLA-A*11:01 and targeted by CD8+ T cells in HLA-A*11:01+ donors.

Conclusions regarding immunodominance are made on very few donors (e.g. 2 out of 4 donors, Fig. 2A; 1 out of 3 donors, Fig. 2B etc.), thus resulting limited and potentially biased due to the low number of donors tested.

Reviewer #2: Main points:

C1R cells express endogenous HLA-C*04:01, so eluted peptides may bind to this allele rather than the transduced HLA-A*11:01. The data is corrected by subtracting peptides identified in previous studies using similar technology. This limitation is clearly mentioned in the manuscript already.

Another limitation concerns the low number of PBMC samples stimulated with a given peptide in screening experiments testing PBMC peptide reactivity, derived from 3 individuals per peptide only. This may have reduced the number of peptides identified, and could also have introduced false positive results, particularly since all 4 indigenous donors were also positive for HLA-A*02:01. Identified peptides were tested with more PBMC samples in the experiment shown in figure 7, but also there the possibility that responses to the identified “HLA-A*11:01” peptides were actually caused by presentation on HLA-A*02:01 is not excluded. This issue could be addressed by stimulating donor PBMCs with the identified peptides presented by C1R cells transduced either with HLA-A*11:01 or -A*02:01.

Also regarding the low number of PBMC samples stimulated with any given peptide, how did the authors define immunodominance for the purpose of this manuscript?

**Part III – Minor Issues: Editorial and Data Presentation Modifications**

Reviewer #1: Plots result often difficult to interpret:

1) Which donors have been used in each experiment in Fig. 2, 4, 5 and 7? In order to help readers interpret the plots, each donor should be shown with a different symbol and the donor code should be indicated in the legend. Did you use the same or different donors for the screenings showed in Fig. 2, 4, 5 and 7?

2) In Fig 5 different symbols are used to distinguish between Indigenous and non-indigenous donors. This reviewer wonders whether this info is really relevant for the interpretation of the results.

3) How did the authors set the cut off of positive or negative response? The results are shown as % IFN-γ+ of CD8+ T cells, but what was the range of basal IFN-γ expression? Showing the “No Ag” condition as well as providing a description of the rationale for determining positivity would be helpful.

4) Information about the antibody staining panels and gating strategy used to identify epitope-reacting CD8+ T cells is not provided in the text. This information should be added in the method section. Did you gate on CD8+ IFN-γ+ cells or only on IFN-γ+ total population? How did you define activated CD8+ T cells when showing the combination of cytokine expression (CD107a, IFN-γ, MIP1-b, TNF)?

5) When possible, statistical analysis should be applied and shown in the plots (e.g. Fig 5).

Reviewer #2: No comments

PLOS authors have the option to publish the peer review history of their article (what does this mean?). If published, this will include your full peer review and any attached files.

Reviewer #1: No

Reviewer #2: No
---

## [Editor Report · Decision Letter 1]

3 Feb 2022

Dear A/Prof Kedzierska,

We are pleased to inform you that your manuscript 'HLA-A*11:01-restricted CD8+ T cell immunity against influenza A and influenza B viruses in Indigenous and non-Indigenous people' has been provisionally accepted for publication in PLOS Pathogens.

Best regards,

Christian Munz

Associate Editor

PLOS Pathogens

Sonja Best

Section Editor

PLOS Pathogens

Kasturi Haldar

Editor-in-Chief

PLOS Pathogens

orcid.org/0000-0001-5065-158X

Michael Malim

Editor-in-Chief

PLOS Pathogens

orcid.org/0000-0002-7699-2064

The authors have significantly improved the manuscript by increasing the number of HLA-A*1101+ tested individuals to 24 (identified after HLA typing of more than 300 donors). They confirm their identified HLA-A*1101 binding peptides and immunogenicity of these. Furthermore, they have now performed more stringent statistical analysis for positive T cell responses and visualized individual donors by separate symbols in the figures. The presented study contributes to our understanding of influenza specific cellular immunity in Asian populations.
---

## [Editor Report · Acceptance letter]

25 Feb 2022

Dear A/Prof Kedzierska,

We are delighted to inform you that your manuscript, "HLA-A*11:01-restricted CD8+ T cell immunity against influenza A and influenza B viruses in Indigenous and non-Indigenous people," has been formally accepted for publication in PLOS Pathogens.

Best regards,

Kasturi Haldar

Editor-in-Chief

PLOS Pathogens

orcid.org/0000-0001-5065-158X

Michael Malim

Editor-in-Chief

PLOS Pathogens

orcid.org/0000-0002-7699-2064